# Alkylamine-tethered molecules recruit FBXO22 for targeted protein degradation

Chrysanthi Kagiou [1], Jose A. Cisneros [1], Jakob Farnung [2], Joanna Liwocha[2], Fabian Offensperger [1], Kevin Dong[3], Ka Yang [3], Gary Tin [1], Christina S. Horstmann[1,4], Matthias Hinterndorfer[1], Joao A. Paulo [3], Natalie S. Scholes [1], Juan Sanchez Avila [1], Michaela Fellner[5], Florian Andersch[5], J. Thomas Hannich [1], Johannes Zuber [5], Stefan Kubicek [1], Steven P. Gygi [3], Brenda A. Schulman [2] & Georg E. Winter [1]✉

Targeted protein degradation (TPD) relies on small molecules to recruit proteins to E3 ligases to induce their ubiquitylation and degradation by the proteasome. Only a few of the approximately 600 human E3 ligases are currently amenable to this strategy. This limits the actionable target space and clinical opportunities and thus establishes the necessity to expand to additional ligases. Here we identify and characterize SP3N, a specific degrader of the prolyl isomerase FKBP12. SP3N features a minimal design, where a known FKBP12 ligand is appended with a flexible alkylamine tail that conveys degradation properties. We found that SP3N is a precursor and that the alkylamine is metabolized to an active aldehyde species that recruits the SCF^FBXO22 ligase for FKBP12 degradation. Target engagement occurs via covalent adduction of Cys326 in the FBXO22 C-terminal domain, which is critical for ternary complex formation, ubiquitylation and degradation. This mechanism is conserved for two recently reported alkylamine-based degraders of NSD2 and XIAP, thus establishing alkylamine tethering and covalent hijacking of FBXO22 as a generalizable TPD strategy.

Targeted Protein Degradation (TPD) has gained significant attention in recent years as a therapeutic modality that promises to overcome the limitations of conventional, inhibitor-centric small-molecule design.

TPD is based on the principle that small molecules can induce molecular proximity between an E3 ubiquitin ligase and a target protein of interest (POI) to trigger POI ubiquitylation and ensuing degradation by the proteasome. On a high level, the field distinguishes two major classes of degraders.

Molecular glue degraders (MGDs) typically function by binding to either the E3 or the POI, modifying the protein surface to induce novel or stabilize existing protein–protein interactions. The resulting ternary complex (POI-MGD-E3) is hence highly cooperative and the MGD orchestrates several protein–protein interactions at the binding interface. The most prominent MGDs are thalidomide and its analogs (commonly referred to as immunomodulatory drugs or IMiDs), which bind to the E3 ligase CRL4^CRBN thereby causing recruitment and degradation of a suite of zinc finger transcription factors[1–4]. Noteworthy, the identification of IMiDs and other MGDs was serendipitous[5], but an increasing toolbox of MGD discovery strategies promises to rationalize future MGD identification[6–11]. The other prominent class of degraders is heterobifunctional proteolysis targeting chimeras (PROTACs)[12]. PROTACs simultaneously bind the POI and the E3 ligase

---

[1]CeMM Research Center for Molecular Medicine of the Austrian Academy of Sciences, 1090 Vienna, Austria. [2]Department of Molecular Machines and Signaling, Max Planck Institute of Biochemistry, Martinsried, Germany. [3]Department of Cell Biology, Harvard Medical School, Boston, MA, USA. [4]St. Anna Children's Cancer Research Institute, Vienna, Austria. [5]Research Institute of Molecular Pathology, Vienna BioCenter, 1030 Vienna, Austria. ✉e-mail: gwinter@cemm.oeaw.ac.at

via dedicated ligands that are connected via a flexible linker. The chimeric nature of PROTACs allows their rational design but overall depends on the availability of well-defined ligands for the POI and the E3.

Around twenty degraders have entered human clinical investigation, all of which function by coopting either CRL4[CRBN] or CRL2[VHL] [13,14]. This establishes the motivation to expand the set of actionable E3 ligases with several objectives in mind[15]. First, accessing new E3 ligases provides a strategy to address resistance mechanisms that are either already clinically observed or expected, particularly in oncology settings[16–19]. Second, VHL and CRBN are, with few exceptions ubiquitously expressed[20], thus preventing tissue- or cell-type selective degradation strategies which would have the potential of an increased therapeutic index. Third, not all POIs are equally amenable to CRBN- or VHL-based degraders, possibly due to incompatibilities in surface topologies and hence an inability to form a productive ternary complex[21–23]. Over the last years, strategies employing electrophilic fragments either as affinity reagents or as putative E3 binders have been successful in unlocking several E3 ligases, including CRL4[DCAF11], CRL4[DCAF16], RNF4, or RNF114 for prototypic PROTAC design, but additional work will be required to understand or realize their translational potential[24–28].

In addition to these two paradigmatic degrader classes, a suite of insufficiently characterized, seemingly "monovalent" degraders directly bind a POI through a well-defined molecular recognition yet induce POI degradation through elusive mechanisms. Functional annotation of these mechanisms promises to reveal novel strategies for degrader design and to unlock additional E3 ligases for chemical exploration. Among others, this has been exemplified by functional studies with the monovalent BRD2/4 degrader GNE-0011 that functionally depends on the CRL4[DCAF16] ligase[29–31]. Similarly, recent reports disclosed "compound 10", a small-molecule degrader of XIAP that consists of a known XIAP binder appended to a flexible primary alkylamine[32]. Initial mechanistic characterization of compound 10 led to a model where the free amine would be directly polyubiquitylated by XIAP *in cis*, thereby inducing its proteasomal degradation[32].

Here, we describe the mechanistic characterization of a serendipitously discovered degrader of the prolyl isomerase FKBP12. Akin to compound 10, it consists of an alkylamine tail that is appended on a target-binding ligand, here the synthetic ligand of FKBP12 (SLF) and is hence termed SLF-PEG3-NH2 (SP3N). To unravel the mechanism of action of SP3N, we coupled FACS-based CRISPR/Cas9 knockout screens with quantitative proteomics and biochemical reconstitutions. This led us to identify that SP3N recruits the SCF[FBXO22] ligase to induce FKBP12 polyubiquitylation and ensuing proteasomal degradation. In accordance with a recently reported alkylamine degrader targeting the histone methyltransferase NSD2[33], we show that SP3N is a precursor that is metabolized into an active aldehyde species. Through targeted mutagenesis, functional reconstitutions, and intact mass spectrometry, we show that the aldehyde adducts a cysteine residue (C326) in the C-terminal domain of FBXO22, and that this covalent engagement is functionally required for ternary complex formation, productive ubiquitylation and degradation. Importantly, our data imply that compound 10, as well as the alkylamine-based degrader of NSD2 functionally converge on the same, PROTAC-like mechanism. Collectively, our data therefore outline a roadmap to rational degrader development and unlocking FBXO22 for TPD applications.

## Results

### SP3N-induced FKBP12 degradation depends on the primary amine and is UPS dependent

En route to the development of a PROTAC candidate library targeting the prolyl isomerase FKBP12, we serendipitously discovered a set of small molecule precursors with unexpected degradation activity. Active precursors merely consist of a free primary alkylamine attached to the synthetic ligand of FKBP12 (SLF) (Supplementary Fig. 1a, b). Among the assayed molecules, attachment of a PEG3-NH2 moiety to SLF (SP3N) yielded the most potent degrader which was thus prioritized for ensuing studies (Fig. 1a). To monitor drug-induced changes in FKBP12 stability, we engineered KBM7 cells to express an FKBP12-BFP-P2A-mCherry reporter compatible with FACS analysis. Leveraging this reporter, we could show that SP3N efficiently degrades FKBP12 in a time- and dose-dependent manner (Fig. 1b). In addition, competition experiments with excess amount of SLF fully rescued from degradation, confirming the requirement of FKBP12 target engagement for productive degradation (Supplementary Fig. 1c). Degradation activity of SP3N was completely abrogated by acetylating the free primary amine (SP3NAc), thus confirming its relevance akin to previously disclosed alkylamine degraders targeting XIAP or NSD2 (Fig. 1a, c, d)[32,34]. To assay the proteome-wide degradation specificity of SP3N, we conducted MS-based whole proteome analysis using tandem mass tags in HEK293T cells. Among the 8958 identified proteins, SP3N selectively degraded FKBP12 after a 16 h incubation. In contrast, the acetylated analog SP3NAc did not prompt destabilization of FKBP12 or any other protein (Fig. 1e, Supplementary Data 1). To gain a better understanding of the mechanism of the SP3N-mediated degradation, we conducted chemical competition experiments with the proteasome inhibitor carfilzomib and the ubiquitin-activating enzyme (UAE) inhibitor TAK243. Both inhibitors fully prevented FKBP12 degradation, indicating a dependency on the ubiquitin-proteasome system (UPS) (Fig. 1f). In addition, pre-treatment with the NEDD8-activating enzyme (NAE) inhibitor MLN4924 fully rescued from degradation, highlighting a functional requirement on NEDD8 conjugation, and hence implying the functional involvement of a Cullin-RING ligase (CRL) in the SP3N-induced degradation of FKBP12[35].

### SP3N recruits FBXO22 for FKBP12 degradation

To identify cellular effectors required for SP3N-induced FKBP12 degradation in an unbiased manner, we employed a FACS-based CRISPR/Cas9 knock-out screen with a UPS-focused sgRNA library (Supplementary Data 2). To this end, we transduced KBM7 cells expressing doxycycline-inducible Cas9 (iCas9) and the FKBP12-BFP-P2A-mCherry reporter with a UPS-focused sgRNA library. Three days post-doxycycline induction cells were treated with SP3N and sorted based on the FKBP12-BFP expression levels into three distinct populations—the highest and lowest 5% of BFP-expressing cells (FKBP12[HIGH] and FKBP12[LOW], respectively), along with the 30% of cells expressing average levels of BFP (FKBP12[MID]). As a control degrader, we used dFKBP1, a previously reported SLF-based PROTAC that is dependent on the CUL4[CRBN] ligase[36]. As expected, FACS-based CRISPR/Cas9 screens of dFKBP1 revealed all components of the CUL4[CRBN] ligase complex alongside subunits of the proteasome as well as the COP9 signalosome in the FKBP12[HIGH] fraction. This implies that CRISPR/Cas9-mediated disruption of these genes abolishes dFKBP1 activity and thus establishes validity of our screening setup (Supplementary Fig. 2a, Supplementary Data 3). Turning our focus to screens treated with SP3N, we identified a profound enrichment of the substrate receptor FBXO22 and other components of the SCF[FBXO22] ligase complex in the FKBP12[HIGH] population, again accompanied by components of the proteasome and COP9 signalosome, thus corroborating chemical competition experiments (Fig. 2a, Supplementary Data 3). To validate the screen results, we proceeded with arrayed gene knockout and reconstitution experiments. Population-level FBXO22 disruption in KBM7 iCas9 cells completely rescued SP3N-induced FKBP12 degradation, as assayed via the FKBP12-BFP-P2A-mCherry stability reporter. In contrast, disruption of the control locus AAVS1 did not affect SP3N efficacy (Fig. 2b, Supplementary Fig. 2b). These results were further corroborated by comparing SP3N-induced FKBP12 degradation in HEK293T wildtype (WT) cells compared to an isogenic, clonal FBXO22 knockout line (FBXO22 KO; Fig. 2c). In addition, reconstitution of

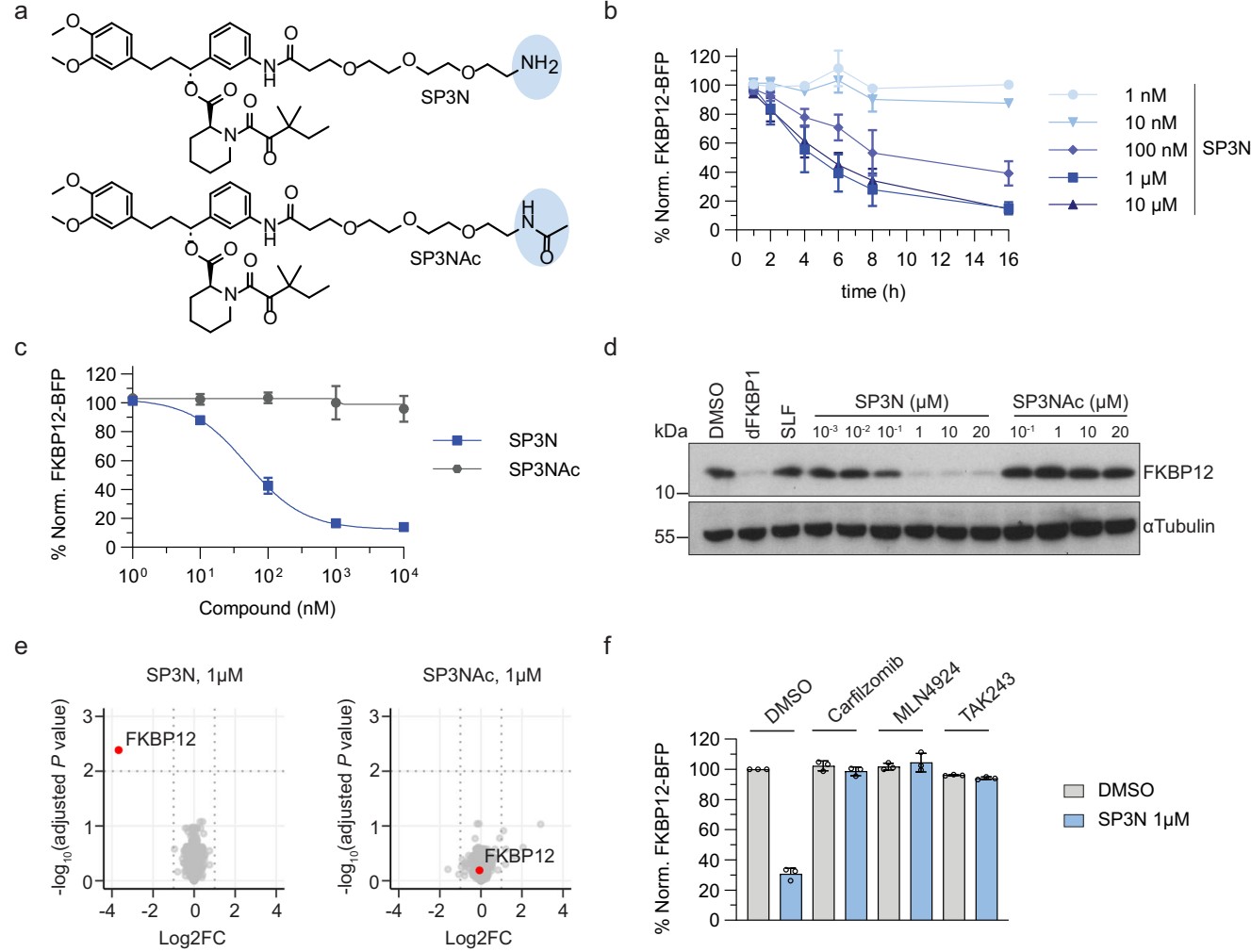

**Fig. 1 | SP3N-induced FKBP12 degradation depends on the primary amine and is UPS-dependent. a** Structures of SP3N and SP3NAc. **b** Flow-cytometry based degradation assay upon SP3N treatment. KBM7 iCas9 cells expressing the FKBP12-BFP-P2A-mCherry reporter were treated with the indicated concentrations of SP3N for 1-16 h. **c** Flow-cytometry based degradation assay in KBM7 iCas9 FKBP12-BFP-P2A-mCherry cells treated with SP3N or SP3NAc for 16 h. **d** Immunoblot of FKBP12 in HEK293T Nluc-3xFlag-FKBP12-NLS cells treated with DMSO, 1 μM dFKBP1, 10 μM SLF and the indicated concentrations of SP3N or SP3NAc for 16 h. αTubulin is the loading control. Representative image of $n = 3$ independent experiments. **e** Whole

proteome analysis using tandem mass tag quantification in HEK293T cells treated with DMSO, 1 μM SP3N or 1 μM SP3NAc for 16 h. Log2 fold-changes (Log2FC) and $-\log_{10}$-transformed Benjamini–Hochberg adjusted one-way analysis of variance (ANOVA) $P$ value compared with DMSO treatment. Data from $n = 3$ replicates. **f** Flow-cytometry based degradation assay in KBM7 iCas9 FKBP12-BFP-P2A-mCherry cells pre-treated with DMSO, 1 μM carfilzomib, 1 μM MLN4924, or 500 nM TAK243 for 1 h before the addition of 1 μM SP3N for 6 h. For all the flow-cytometry based degradation assays (**b**, **c**, **f**), the BFP/mCherry ratio was normalized to DMSO and the data shown are from $n = 3$ biological replicates, mean ± s.d.

FBXO22 KO cells with FBXO22 cDNA restored degradation to levels comparable to WT cells (Fig. 2c). Finally, depletion of the FBXO22 adaptor protein SKP1 also rescued from SP3N-mediated FKBP12 degradation, further confirming that the SCF$^{FBXO22}$ complex is required for the degradation process (Supplementary Fig. 2c).

To assess if SP3N induces proximity between FBXO22 and FKBP12, we performed co-immunoprecipitation experiments of 2HA-tagged FBXO22 and 3xFlag-tagged FKBP12. As expected, in the input fraction SP3N treatment induced FKBP12 degradation, which was abolished with carfilzomib treatment. In the IP fraction we observed an interaction of FKBP12 with FBXO22 exclusively upon treatment with SP3N in a dose-dependent manner (Fig. 2d). In line with a lack of SP3NAc-induced degradation, we did not observe an interaction between 2HA-FBXO22 and 3xFlag-FKBP12 upon cellular treatment with SP3NAc (Fig. 2d). To further corroborate these results, we developed a proximity assay that is compatible with measurements in intact cells. To this end, we turned to a nanoluciferase complementation assay (NanoBiT®) capable of assessing ternary complex formation between FBXO22 and FKBP12 (Fig. 2e). HEK293T cells were co-transfected with LgBiT-FKBP12

and SmBiT-FBXO22 followed by treatment with DMSO, SP3N or SP3NAc. Supporting co-IP data, we observed a pronounced and dose-proportional increase of bioluminescence after cellular SP3N treatment, indicative of drug-induced ternary complex formation (Fig. 2f). Noteworthy, the less potent degrader SP2N also induced FKBP12-FBXO22 complex formation, albeit at a lower magnitude (Supplementary Fig. 2d). As expected, SP3NAc treatment did not induce a bioluminescence signal (Fig. 2f). Competition of SP3N with excess SLF blocked luminescence induction, thus further corroborating the requirement for direct FKBP12 engagement (Fig. 2g). In conclusion, these data support a mechanism whereby SP3N induces proximity between FKBP12 and FBXO22 to prompt FKBP12 degradation by FBXO22.

## SP3N is a precursor metabolized to an active aldehyde species

Since primary alkylamines represent a potential metabolic liability, we sought to identify if SP3N might be converted into a functionally relevant metabolite. Of note, a similar mechanism has recently been disclosed for UNC8732, an alkylamine-based degrader targeting the

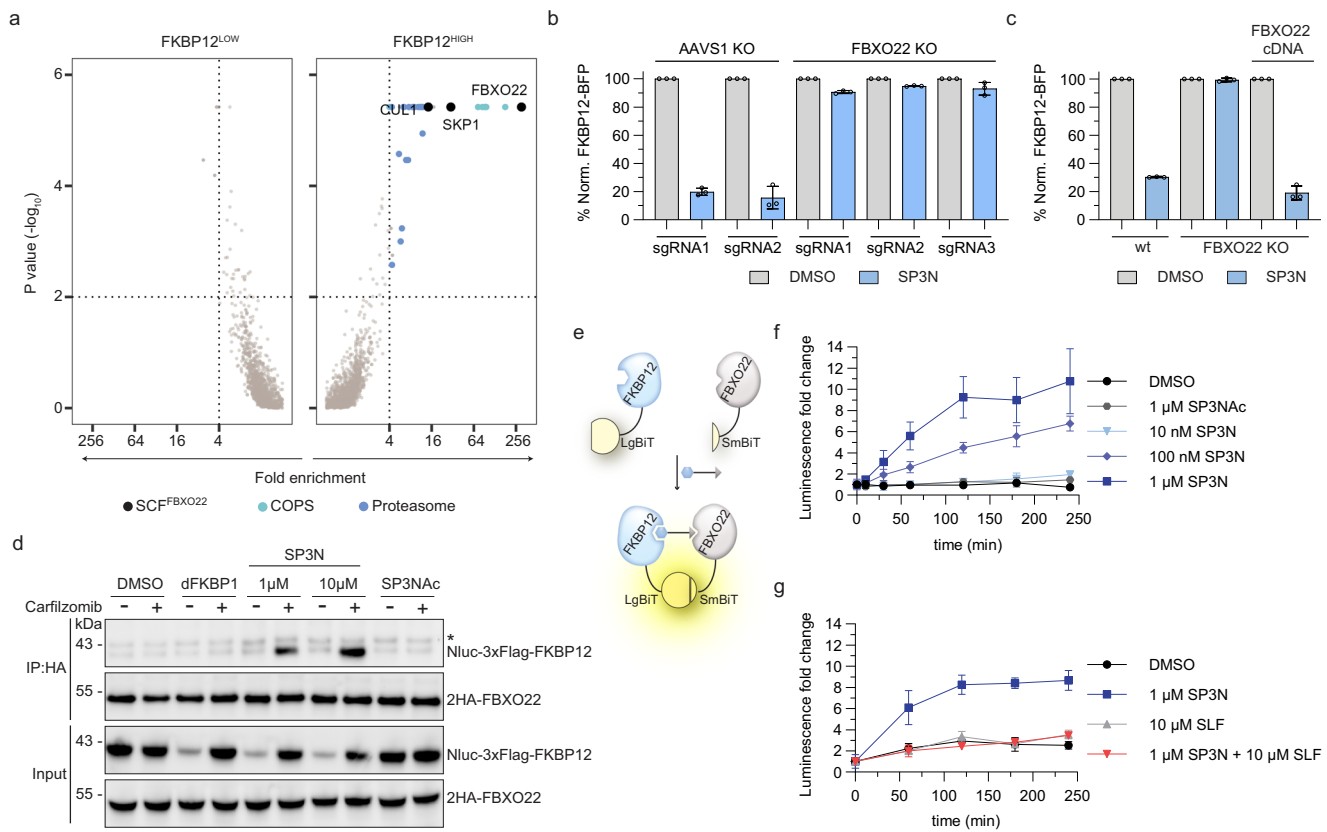

**Fig. 2 | SP3N recruits FBXO22 for degradation. a** FACS based CRISPR/Cas9 stability screen. KBM7 iCas9-FKBP12-BFP-P2A-mCherry reporter cells were transduced with a UPS-focused sgRNA library, treated with DMSO or 250 nM SP3N for 16 h and sorted based on the FKBP12-BFP levels. Fold changes and p-values of the FKBP12^HIGH and FKBP12^LOW populations were calculated by comparison with the FKBP12^MID population using two-sided negative binomial test (MAGeCK). Significant hits: fold-enrichment ≥ 4 and -log$_{10}$ Pvalues ≥ 2 (dotted lines). Cyan: COP9 signalosome; blue: proteasome subunits; black: SKP1-CUL1-FBXO22 complex. Data from $n = 2$ replicates. **b** Flow cytometry-based degradation assay for screen validation. KBM7 iCas9-FKBP12-BFP-P2A-mCherry cells were transduced with sgRNAs targeting the control locus AAVS1 or FBXO22 and treated with DMSO or 1 µM SP3N for 16 h. **c** Flow cytometry-based degradation assay for FBXO22 reconstitution. FBXO22 clonal knock-out of HEK293T FKBP12-BFP-P2A-mCherry cells were reconstituted with the 2HA-FBXO22 cDNA and treated with DMSO or 1 µM SP3N for 16 h. For (**b**, **c**) the BFP/mCherry ratio was normalized to DMSO and the data is the mean ± s.d of $n = 3$ biological replicates. **d** Co-immunoprecipitation of 2HA-FBXO22 and Nluc-3xFlag-FKBP12 following treatment with DMSO, 1 µM dFKBP1, 1 µM or 10 µM SP3N or 1 µM SP3NAc for 4 h in the presence of 1 µM carfilzomib. Samples without carfilzomib were used as controls. IP immunoprecipitation. *unspecific band. Representative image of $n = 3$ independent experiments. **e** Schematic representation of the NanoLuc® Binary Technology (NanoBiT) assay. Upon treatment with molecules that induce proximity of the N-terminal LgBiT-FKBP12 and N-terminal SmBiT-FBXO22 fusions, an active Nluc enzyme is formed that can generate luminescence. **f, g** NanoBiT® assays in HEK293T co-transfected with LgBiT-FKBP12 and SmBiT-FBXO22 fusions, pre-treated with 1 µM carfilzomib and Vivazine substrate before treatment with the indicated compounds. Fold change of luminescence is normalized to timepoint 0, right before the treatment. Mean ± s.d of $n = 3$ technical replicates; representative of $n = 3$ biological replicates. **f** Dose–response for complex formation. Luminescence was monitored after 5 min, 10 min, 30 min and every hour up to 4 h, post-treatment. **g** Competition assay with SLF. Luminescence was monitored every hour up to 4 h, post-treatment.

histone methyltransferase NSD2[33]. In brief, Nie et al. could demonstrate that UNC8732 acts as a precursor that is metabolized to a potent aldehyde by amine oxidases present in the fetal calf serum (FCS) in the cell culture media. To test if SP3N follows a similar mechanism (Fig. 3a), we treated cells with SP3N in full media containing 10% FCS or in Opti-MEM without FCS, for 8 h and tested degradation. Indeed, FKBP12 degradation only occurred in media with FCS, suggesting that SP3N might undergo a similar metabolic conversion towards an active species (Fig. 3b). To confirm the presence of the SP3N-derived aldehyde (SP3CHO) and the dependence on FCS for this metabolic step, we treated KBM7 cells with SP3N in IMDM + 10% FCS or Opti-MEM without FCS and used ultra-performance liquid chromatography-mass spectrometry (UPLC-MS/MS) to detect the formation of the aldehyde species. Our results reveal the detection of SP3CHO already at 6 h of incubation, and only in conditions containing FCS (Fig. 3c). Taken together, these results indicate that SP3N undergoes an FCS-dependent metabolic step towards an active aldehyde species. The requirement of FCS for the activity of the alkylamine degrader insinuates the involvement of extracellular amine oxidases in the

conversion of SP3N to SP3CHO. To further test this hypothesis, we incubated SP3N with recombinant porcine diamine oxidase (DAO) and subsequently treated KBM7 cells with this solution in media without FCS. Supporting the notion of an involvement of diamine oxidases in the metabolic conversion, DAO pre-treatment of SP3N was sufficient to convert it into an active degrader (Fig. 3d). We next performed a time-course treatment of SP3N with DAO and used UPLC-MS/MS to quantify SP3CHO levels. Confirming the involvement of DAO in this metabolic conversion, we quantified increasing amounts of SP3CHO over time and only upon treatment with the enzyme (Supplementary Fig. 3a).

To confirm that the aldehyde is the active species, we synthesized SP3CHO, as well as the hydrolytically labile, protected aldehyde bisulfite adduct of SP2N (SP2CHOp; given technical challenges in directly synthesizing SP2CHO). In cellular degradation assays, in the presence of FCS, both aldehyde species outperform their matched alkylamine analog (Fig. 3e, Supplementary Fig. 3b). This suggests that metabolic conversion might act as a rate-limiting step (Fig. 3c). Moreover, in contrast to SP3N and SP2N, FKBP12 degradation induced by SP3CHO and SP2CHOp is independent of FCS (Fig. 3e,

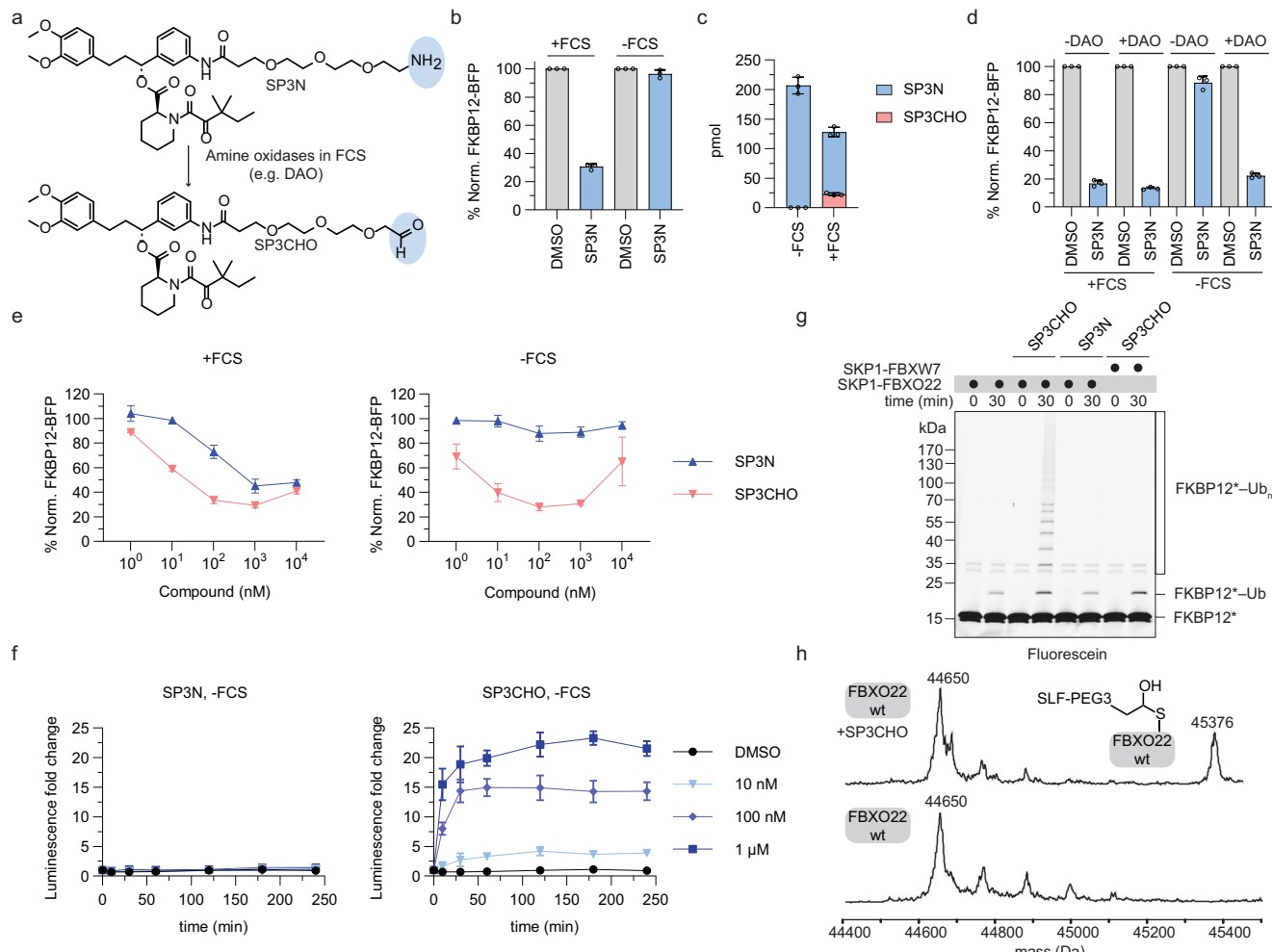

**Fig. 3 | SP3N is a precursor metabolized to an active aldehyde. a** Structures of SP3N and the SP3CHO metabolite. DAO diamine oxidase. **b** Flow-cytometry based degradation assay. KBM7 iCas9 FKBP12-BFP-P2A-mCherry reporter cells were treated with DMSO or 1 μM SP3N in IMDM + 10% FCS or Opti-MEM - FCS for 8 h. **c** Quantification of SP3N and SP3CHO (pmol) using UPLC-MS/MS, in KBM7 iCas9 cells. 1 μM SP3N was added in IMDM + 10 % FCS or Opti-MEM - FCS and incubated at 37 °C for 6 h. Mean ± s.d of $n = 3$ technical replicates. **d** Flow-cytometry based degradation assay with porcine diamine oxidase (DAO). 10 μM of SP3N were incubated with 40 μg DAO in PBS for 4 h. KBM7 iCas9 FKBP12-BFP-P2A-mCherry reporter cells were treated with the pre-incubated solution in Opti-MEM + 10% FCS or Opti-MEM − FCS, for 16 h. **e** Flow-cytometry based degradation assay in KBM7 iCas9 FKBP12-BFP-P2A-mCherry reporter cells in IMDM + 10% FCS or Opti-MEM - FCS treated with DMSO or the indicated concentrations of SP3N or SP3CHO for 6 h. **f** NanoBiT assay as described in Fig. 2e. Before treatment, the cells were washed

with PBS and the indicated concentrations of SP3N or SP3CHO were added to the cells in Opti-MEM - FCS. The luminescence was monitored after 10 min, 30 min and every hour up to 4 h post-treatment and normalized to timepoint 0. Mean ± s.d of $n = 3$ technical replicates; representative plot of $n = 3$ biological replicates. **g** In vitro ubiquitylation assay of fluorescently labeled FKBP12 with activated (i.e. neddylated) SCF$^{FBXO22}$ in the presence of 10 μM SP3N or SP3CHO. The neddylated SCF$^{FBXW7}$ is used as negative control. Representative image of $n = 2$ biological replicates. **h** Intact mass spectrometry for the identification of FBXO22-SP3CHO complex formation. 20 μM FBXO22-SKP1 complex were incubated with 100 μM SP3CHO for 10 min and analyzed with LC-MS. The spectra of FBXO22 and FBXO22-SP3CHO are shown; $n = 2$ biological replicates. Expected mass for FBXO22:44652 Da and for FBXO22-SP3CHO adduct: 45374 Da. For the flow cytometry-based degradation assays (**b**, **d**, **e**), the BFP/mCherry ratio was normalized to DMSO and the data is the mean ± s.d of $n = 3$ biological replicates.

Supplementary Fig. 3b). Interestingly, SP3CHO is slightly more potent and shows a more pronounced hook effect (a phenomenon typically observed with PROTACs) in the absence of FCS (Fig. 3e). UPLC-MS/MS based, targeted quantification of SP3CHO levels after short cellular treatment revealed significantly elevated levels of SP3CHO in cells treated in media lacking FCS as a plausible mechanism for this differential potency (Supplementary Fig. 3c). Moreover, as our data indicates that the aldehyde metabolite mediates the degradation, we sought to confirm that it also induces proximity between FBXO22 and FKBP12, independent of FCS availability. To this end, we employed the aforementioned NanoBiT® assay and monitored real-time complex formation. In line with our hypothesis, our findings revealed that SP3CHO induces dose-dependent ternary complex formation irrespective of FCS availability. In contrast, SP3N failed to induce

interactions in FCS-deprived media, further supporting that SP3CHO is the active SP3N metabolite (Fig. 3f, Supplementary Fig. 3d).

To reconstitute the proposed mechanism in vitro, we purified recombinantly expressed activated (i.e. neddylated) SCF$^{FBXO22}$ (Supplementary Fig. 3e), and FKBP12, and performed ubiquitylation assays with or without compound treatment. In support of a precursor mechanism of action, SP3N treatment was insufficient to induce polyubiquitylation of FKBP12. In contrast, SP3CHO treatment prompted clear poly-ubiquitylation of FKBP12 by SCF$^{FBXO22}$, but not by an unrelated CRL E3 ligase (neddylated SCF$^{FBXW7}$) (Fig. 3g). Similar results were observed with SP2CHOp, which induced FKBP12 poly-ubiquitylation, while SP2N did not show any effects in vitro (Supplementary Fig. 3f). Given the electrophilic nature of SP3CHO and considering recent findings with the aforementioned NSD2 degrader UNC8732, we surmised that SP3CHO can form an adduct with FBXO22 via a covalent

and reversible hemithioacetal. Indeed, intact mass spectrometry clearly revealed a mass corresponding to the SP3CHO-adducted FBXO22 (Fig. 3h). Having established that SP3N covalently adducts FBXO22, and based on another recent report that established SCF[FBXO22] as a ligase that can be harnessed with a covalent, chloroacetamide containing PROTAC, we wanted to investigate if we could replace the aldehyde with alternative warheads and synthesized four additional electrophilic compounds, namely the SP3-chloroacetamide, the SP3-acrylamide and the respective SP2-based analogs[37]. Interestingly, none of these compounds exhibited robust FKBP12 degradation, indicating that these SLF-based aldehydes are favored over the other SLF-based electrophiles in inducing FBXO22-depedent protein degradation (Supplementary Fig. 3g).

## FBXO22 is recruited through its C326 for degradation

Given the reactivity of the active aldehyde species, we next wanted to assess a possible covalent engagement on FBXO22. To map the functionally required and covalently engaged cysteine residue in FBXO22, we mutated five cysteine residues in the C-terminal region of FBXO22 (amino acids 143-365) that has been reported to play a role in substrate binding[38,39]. To this end, we turned to genetic reconstitution experiments where we re-introduce FBXO22 cysteine mutants in FBXO22 knockout cells and assess their effect on SP3N-induced FKBP12 degradation utilizing the aforementioned FKBP12-BFP-P2A-mCherry stability reporter. SP3N- or SP3CHO-induced degradation was maintained by all FBXO22 mutations with the exception of C326A (Fig. 4a, b, Supplementary Fig. 4a, b). To rule out potential deleterious effects of the C326A mutation on FBXO22, we turned to Nano differential scanning fluorimetry (NanoDSF) which revealed thermal stability (Supplementary Fig. 4c). In addition, FBXO22 WT and C326A are similarly sensitive to pharmacologically induced auto-degradation via COP9 signalosome inhibition (Supplementary Fig. 4d), supporting the notion that the mutant incorporates into a functional SCF complex also in cells[40–42]. Further, an in vitro autoubiquitylation assay demonstrated comparable autoubiquitylation of recombinant FBXO22-WT and FBXO22-C326A variants, hence again suggesting intact SCF complex activity (Supplementary Fig. 4e). To orthogonally confirm the relevance of C326, we employed NanoBiT® assays with SmBiT-FBXO22-WT, SmBiT-FBXO22-C326A or SmBiT-FBXO22-C228A mutants. While SP3CHO induced comparable bioluminescence levels for FBXO22-WT and the C228A negative control mutant, no induction of bioluminescence and hence no evidence for ternary complex formation could be observed when assaying the C326A mutant (Fig. 4c, Supplementary Fig. 4f). Further corroborating that C326 is essential for the molecular recognition of SP3CHO, SCF[FBXO22-C326] failed to induce ubiquitylation on FKBP12 in in vitro assays (Fig. 4d). Likewise, intact MS analysis of SKP1-FBXO22-C326A treated with SP3CHO revealed no evidence of adduct formation (Fig. 4e). To confirm the proteome-wide selective engagement by SP3CHO, we performed global reactive cysteine profiling in HEK293T cell lysates by TMT-ABPP[43,44]. This revealed that approximately 30% of FBXO22-C326 was engaged by SP3CHO. No other FBXO22 Cys residues were detected as engaging SP3CHO (Supplementary Fig. 4g, h, Supplementary Data 4). Notably, SP3CHO generally exhibited low reactivity with 5 other Cys (HDAC1-C100, GPX4-C102, TARS1-C254, PSMB1-C82;89 and PPAT-C503). Taken together, our results reveal C326 as the site of covalent binding by SP3CHO that is functionally required for ternary complex formation, ubiquitylation and degradation.

## Alkylamine-based degraders functionally depend on FBXO22

To explore the potential generalizability of exploiting FBXO22 for targeted protein degradation, we extended our experiments to two recently reported primary alkylamine-tethered degraders targeting either NSD2 or XIAP, as well as a set of alkylamine-tethered analogues targeting BRD4 (Fig. 4f and Supplementary Fig. 5a)[32–34]. Supporting a

general role of FBXO22 in the mechanism of action of alkylamine-based degraders, no target degradation was observed in FBXO22 KO cells. In contrast, reconstitution with FBXO22-WT re-sensitized KO cells to target destabilization by the NSD2, XIAP and FKBP12 targeting degraders. Notably, reconstitution with FBXO22-C326A failed to re-establish target degradability, suggesting a shared functional dependency on this key residue. Together, these findings support the broader applicability of a TPD strategy whereby target-binding ligands can be equipped with flexible alkylamines to recruit the SCF[FBXO22] ligase for target ubiquitylation and ensuing degradation by the proteasome. Interestingly, a set of alkylamine-based analogues building off the BET-bromodomain inhibitor JQ1 did not degrade BRD4, a target that is frequently utilized as proof of concept for prototypical degraders that co-opt novel E3 ligases (Supplementary Fig. 5a). While immunoblot analysis of the BRD4 transcriptional target MYC implies cellular target engagement of this set of analogues, co-IP experiments reveal a lack of ternary complex formation as a likely reason for the observed lack of degradation (Supplementary Fig. 5 b, c). In sum, these data suggest that the concept of alkylamine-base degraders is generalizable yet will require optimization on a target-by-target level.

## Discussion

Here, we report the serendipitous identification of SP3N, a degrader of the prolyl isomerase FKBP12, which features a minimal design where the known FKBP12 ligand SLF is equipped with an alkylamine extension that conveys the observed degradation properties. Orthogonal mechanistic characterization via quantitative proteomics, functional genomics and biochemical reconstitutions led us to identify that SP3N-induced degradation is highly specific and depends on the recruitment of the SCF[FBXO22] ligase. Further, we employ metabolomics to reveal that SP3N is a precursor that is metabolically converted into an active aldehyde species (SP3CHO) via amine oxidases. Coupling targeted mutagenesis studies with genetic rescue experiments, we identify C326 in the C-terminal putative substrate binding domain of FBXO22 as critical for SP3N/SP3CHO-induced FKBP12 degradation. Further corroborating the critical role of C326, we observe that mutating C326 completely abrogates drug-induced proximity of FKBP12 and FBXO22 in intact cells. Likewise, recombinant FBXO22-C326A is incapable of inducing FKBP12 polyubiquitylation. Lastly, we could extend our findings around SP3N to two recently reported degraders of XIAP and NSD2 that feature a similar alkylamine design[32–34]. Akin to SP3N, both compounds require FBXO22 for target degradation and appear similarly dependent on C326 in this process. Of note, the NSD2-based degrader UNC8732 has also been reported to be subject to metabolization into an aldehyde species[33]. We thus surmise that this metabolic conversion of primary amine tethered precursors into reactive aldehyde species is a general phenomenon.

Expanding the reach of targeted protein degradation to additional E3 ligases has recently been a very active area of research. Fragment-based chemoproteomics as well as focused mechanism of action campaigns have unlocked several E3 ligases, including DCAF11, DCAF16, RNF4 or RNF114 for chemical exploration[24–27,31,45]. Nevertheless, less than 3% of the around 600 E3 ligases encoded in the human genome can be coopted with small-molecule ligands. Among the successfully liganded E3 ligases, a critical evaluation will be required to understand if the identified chemical matter can be progressed towards selective compounds that fulfill probe-like criteria[46]. Likewise, further research will be required to understand which of those E3 ligases provides a tangible differentiation from CRBN and VHL, the two E3 ligases that are harnessed by clinically approved or evaluated MGDs and PROTACs. Aside from potentially addressing and overcoming resistance mechanisms to CRBN- and VHL-based degraders[16–19,47], or from expanding the toolbox of degron-tag approaches[48–50], the disclosed mechanism of FBXO22 dependent,

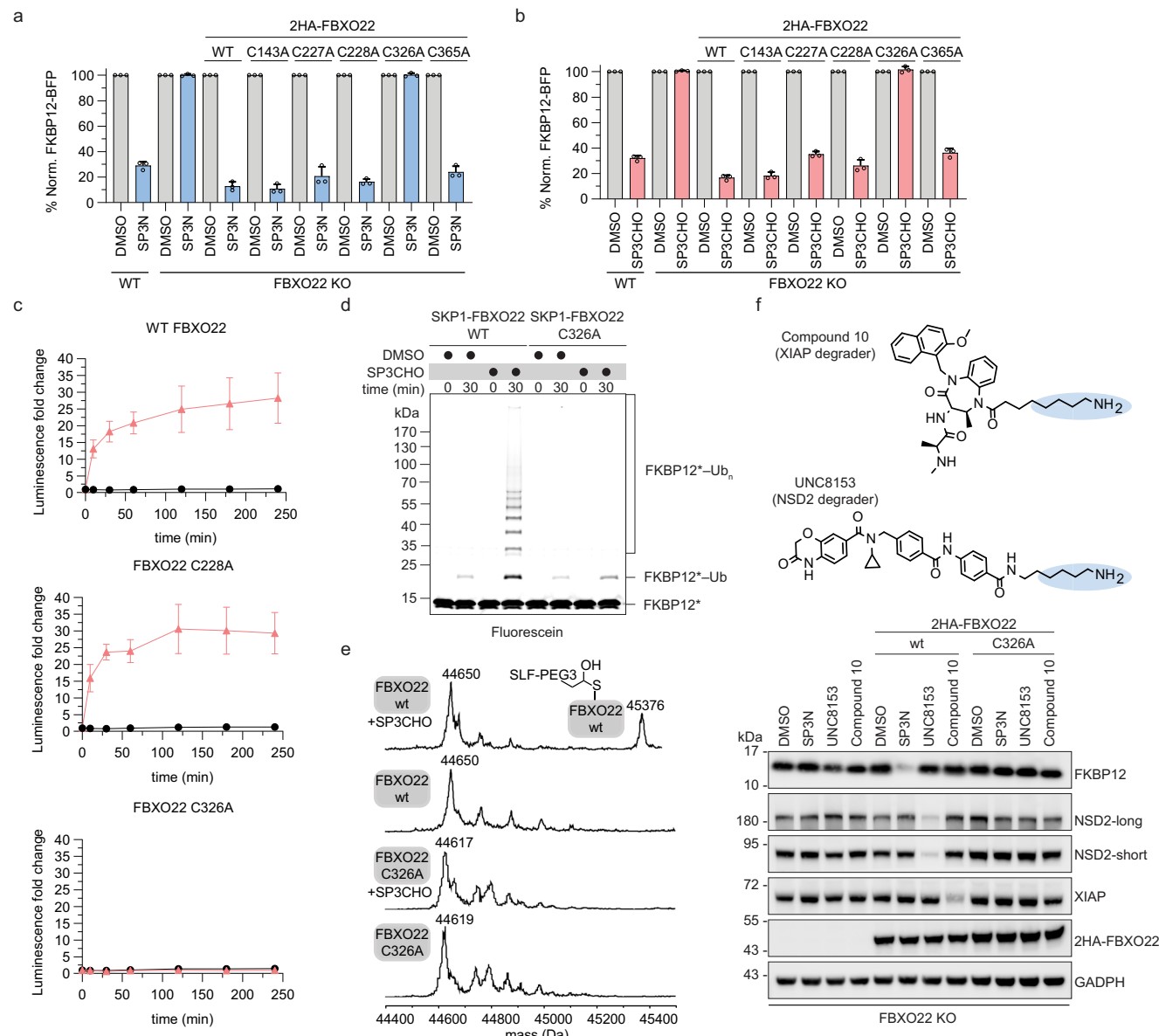

**Fig. 4 | FBXO22-C326 is crucial for the degradation induced by SP3N and other alkylamine-tethered degraders. a**, **b** Flow-cytometry based degradation assays in HEK293T FKBP12-BFP-P2A-mCherry single FBXO22 KO clone transduced with WT, C143A, C227A, C228A, C326A or C365A mutant 2HA-FBXO22 cDNAs. Cells were treated with DMSO or 1 μM SP3N for 16 h (**a**) and with DMSO or 1 μM SP3CHO for 8 h (**b**). Untransduced WT or FBXO22 KO cells were used as controls. The BFP/mCherry ratio was normalized to DMSO for each cell line. Mean ± s.d of $n = 3$ biological replicates. **c** NanoBiT assay in HEK293T cells co-transfected with LgBiT-FKBP12 and SmBiT-FBXO22-WT, SmBiT-FBXO22-C228A or SmBiT-FBXO22-C326A and treated with DMSO or 1 μM SP3CHO, in Opti-MEM - FCS. The luminescence was monitored after 10 min, 30 min and every hour up to 4 h post-treatment. Mean ± s.d of $n = 3$ technical replicates; representative plot of $n = 3$ biological replicates. **d** In vitro ubiquitylation assay of fluorescently labeled FKBP12 with SCF$^{FBXO22-WT}$

or SCF$^{FBXO22-C326A}$ in the presence of DMSO or 10 μM SP3CHO. Representative image of $n = 2$ biological replicates. **e** Intact mass spectrometry of FBXO22-WT versus FBXO22-C326A for SP3CHO binding. 20 μM SKP1-FBXO22-WT or SKP1-FBXO22-C326A were incubated with 100 μM SP3CHO for 10 min and analyzed with LC-MS. The spectra of SKP1-FBXO22-WT −/+ SP3CHO or SKP1-FBXO22-C326A −/+ SP3CHO are shown. Expected mass for FBXO22: 44652 Da. Expected mass for FBXO22-SP3CHO adduct: 45374 Da. Expected mass for FBXO22 C326A: 44619 Da. **f** Immunoblot of HEK293T FKBP12-BFP-P2A-mCherry FBXO22 KO untransduced cells or cells transduced with the 2HA-FBXO22-WT cDNA or 2HA-FBXO22-C326A cDNA followed by treatment with DMSO, 5 μM SP3N, 5 μM UNC8153 (NSD2 degrader) or 5 μM Compound 10 (XIAP degrader) for 16 h. GADPH is the loading control. Western blots are representative of $n = 3$ biological replicates.

alkylamine-based degraders offers some points of differentiation that warrant further exploration.

First, FBXO22 has a well-established role in carcinogenesis and its expression is associated with poor survival in several cancer types[51]. While it is broadly expressed in most tissues, reports of increased levels of expression in certain cancer types, such as lung adenocarcinoma or ovarian cancers, might enable increased degradation efficiencies specifically in malignant cells, which could result in an

expanded therapeutic index[52,53]. This is further supported by TCGA data that indicates the elevated expression levels in tumor tissue as a differentiating characteristic compared to CRBN, the most-frequently pursued E3 ligase for TPD applications (Supplementary Fig. 5d, e). Second, the required precursor conversion is another layer of tumor specificity that could be exploited. Certain cancer types such as colorectal cancer or hepatocellular carcinoma exhibit upregulation of specific amine oxidases, which could enable elevated rates of degrader

precursor conversion in tumors, even though general metabolic liabilities of primary amines would need to be considered[54–56]. Future studies should explore experimental approaches that better reflect the physiological context in which amine oxidases function. One potential avenue would be the use of ex vivo or in vivo models that better mimic the tumor microenvironment and allow for the assessment of tumor-restricted amine-to-aldehyde conversion and target degradation.

Our data, alongside the data reported in Nie et al. clearly highlight an essential functional role of the solvent-accessible Cys326 in the C-terminal domain of FBXO22[33,57,58]. However, future research including a full structural elucidation will be required to dissect the determinants of molecular recognition of the highly flexible alkylamine tail. Moreover, structural understanding will be key to empower rational ligand optimization. This is of particular interest in light of recent findings by Basu et al. highlighting that additional cysteine residues (C227, C228) of FBXO22 are in principle ligandable with chloroacetamide-based compounds[37]. In addition, a more granular understanding of the underpinning molecular recognition will be instrumental to explain limitations in the generalizability of the alkylamine degrader approach. For instance, while alkylamine-based degraders could be leveraged against FKBP12, NSD2, and XIAP, an informer set of alkylamines conjugated to the BET bromodomain ligand JQ1 failed to induce proximity and ensuing degradation of the BET protein BRD4, a target that is otherwise frequently utilized for degrader proof of concept studies.

In conclusion, data presented in this manuscript, together with corroborating evidence from other studies, highlight alkylamine conjugation as a strategy to develop small-molecule precursor degraders that mechanistically converge on harnessing the SCF$^{FBXO22}$ ligase and harbor the potential to be active against a broad spectrum of targets. As such, we expect the presented data to establish motivation for focused FBXO22 ligand discovery and degrader optimization efforts as well as for understanding a putative physiological relevance of reprogramming FBXO22.

## Methods

### Cell culture
KBM7 inducible Cas9 (iCas9) cells (gift from Johannes Zuber / IMP - Research Institute of Molecular Pathology) were cultured in IMDM (Gibco, 21980032) and 293T cells (ATCC, CRL-3216) or Lenti-X 293T (Clontech, 632180) in DMEM (Gibco, 41965062), both supplemented with 100 U ml$^{-1}$ penicillin/streptomycin (Sigma-Aldrich, P4333) and 10% fetal calf serum (FCS, Gibco, A5256701), unless specified otherwise. Cells were grown at 37 °C and 5% $CO_2$ humidified incubator.

### Plasmids/oligonucleotides
For the engineering of the FKBP12-BFP-P2A-mCherry reporter, FKBP12 (Twist Biosciences) was cloned into a pRRL lentiviral vector containing a 3xV5-mTagBFP coupled to mCherry with a P2A self-cleaving peptide for normalization. The BRD4short(s)-BFP-P2A-mCherry stability reporter used in this study has been previously published[31,59]. For the cloning of pLenti6.2-Nluc-3xFlag-FKBP12 or 2HA-FBXO22, 3xFlag-FKBP12-(wt or NLS) (Twist Biosciences) or FBXO22 with synonymous mutations in the PAM and seed sequences (Twist Biosciences) were cloned into the pLenti6.2-Nanoluc-ccdB (Addgene, #87078) or the pLEX-2HA-P2A-puro, using Gateway™ (BP Clonase II, 11789020 and LR Clonase™ II, 11791100 both from Invitrogen) and according to the manufacturer's protocol. Cloning of the pBiT2.1-SmBiT-FBXO22 and pBiT1.1-N-LgBiT-FKBP12 was achieved by restriction enzyme-based cloning. Briefly, FBXO22 or FKBP12 were PCR amplified (Q5 DNA polymerase, NEB, B9027S) from the 2HA-FBXO22 or the pLenti6.2-Nluc-3xFlag-FKBP12 plasmids, respectively, using primers with appropriate restriction enzyme sites. The amplified FBXO22 or FKBP12 fragments were inserted in the restriction enzyme-digested pBiT2.1-N-SmBiT or pBiT1.1-N-LgBiT (Promega, N2014) vectors and ligated using

T4 DNA Ligase (NEB, M0202S) according to the manufacturer's protocol. For the cysteine-mutant 2HA-FBXO22 and SmBiT-FBXO22 plasmids, the 2HA-FBXO22-wt or SmBiT-FBXO22-wt plasmids were mutated using Q5 site-directed mutagenesis (New England Biolabs, E0552), according to the manufacturer's protocol and using oligonucleotides designed with NEBaseChanger (v2.4.2). All plasmids and oligonucleotides/primers used in this study are shown in Supplementary Tables 1 and 2, respectively, and the UPS-focused sgRNA library used for the FACS-based CRISPR/Cas9 stability screen is shown in Supplementary Data 2.

### Compounds
The inhibitor carfilzomib (Cay17554) is from Biomol and JQ1 (1268524-70-4) is from AmBeed. The inhibitors MLN4924 (HY-70062) and TAK243 (HY-100487) and the degraders dFKBP1 (HY-103634), Compound 10 (XIAP degrader-1, HY-115865) and dBet6 (HY-112588) were all purchased from MedChemExpress.

### Virus production and transductions
Lenti-X 293T cells at 70-90% confluency were transfected with the desired lentiviral plasmids and the packaging plasmids (pCMVR8.74 helper, Addgene #22036 and pMD2.G envelope, Addgene #12259) using polyethylenimine (PEI MAX® MW 40,000, Polysciences, 24765). The virus was collected and clarified using 0.45 μm Whatman Puradisc Syringe Filter (cytiva, WH6756-2504). Different dilutions of viral suspension were added to the cells and the cell/virus suspension was centrifuged at 900 g for 45 min and 33 °C.

### Clonal FBXO22 knock-out cell line
To generate clonal FBXO22 knock-out cell line, HEK293T FKBP12-BFP-P2A-mCherry cells were transduced with plasmids expressing the sgRNA 'GATCCAGGTTACGCTCCGAT' targeting FBXO22. After G418 (Sigma-Aldrich, A1720) selection, the cells were transfected with pSpCas9(BB)−2A-Puro (PX459) v2.0 (Addgene, #62988) plasmid using PEI. 48 h post-transfection, single clones were seeded in 96 well plates using CytoFLEX LX sorter (Beckman Coulter) and grown for 2 weeks at 37 °C and 5% $CO_2$ humidified incubator. Several single clones were screened for FBXO22 protein levels using western blot assay and validated with functional rescue assays with the SP3N degrader.

### Flow-cytometry-based degradation assays
KBM7 iCas9 or HEK293T cells expressing FKBP12-BFP-P2A-mCherry reporter or BRD4short-BFP-P2A-mCherry reporter were seeded in 24 well plates at seeding densities of 0.5 or 0.25 × 10$^6$ cells ml$^{-1}$, respectively. The working dilutions of compounds were prepared freshly in media, using 1000x stock solutions in DMSO (Sigma-Aldrich, D1435). For competition experiments the cells were pre-treated with 1 μM Carfilzomib, 1 μM MLN4924 or 500 nM TAK243 for 1 h before the addition of SP3N or co-treated with different concentrations of SLF and SP3N. Post-treatment, cells were collected into Falcon® 5 ml Round Bottom Polystyrene Test Tubes (Corning, 352052) and directly measured with BD LSRFortessa™ Cell Analyzer (BD Biosciences).

### Western blot analysis
Cells, post-treatment with the compounds described in the figure legends, were collected in ice-cold Dulbecco's phosphate-buffered saline (PBS, Gibco, 14190144), washed 2x with PBS and lysed with RIPA buffer (150 mM NaCl, 1% TritonX-100, 0.5% Sodium deoxycholate, 0.1% Sodium dodecyl sulfate, 50 mM Tris pH 8) freshly supplemented with benzonase Nuclease (Merck Millipore, 70746) and Halt™ Protease Inhibitor Cocktail, EDTA-Free (100X) (Thermo Fisher Scientific, 78425). The protein concentration was determined using the Pierce™ BCA Protein Assay (Thermo Fisher Scientific, 23225) and 20 μg of lysate with 4X Bolt™ LDS Sample Buffer (Thermo Fisher Scientific, B0007) supplemented with 10% 2-Mercaptoethanol (Sigma-Aldrich, M3148)

was loaded per lane of NuPAGE 4-12% bis-tris gels (Invitrogen, NP0329PK2). Proteins were transferred to nitrocellulose membranes (Cytiva, 10600002), blocked at room temperature (RT) with 5% dry non-fat milk in Tris-buffered saline-Tween-20 (TBS-T, 50 mM Tris-Cl, pH 7.5, 150 mM NaCl, 0.1% Tween-20) and incubated with primary antibodies overnight at 4 °C. The following day, the membranes were incubated with HRP-conjugated secondary antibodies for 1 h at RT. The membranes were imaged with ChemiDoc™ Touch Imaging System-system (Bio-Rad), using ECL (Amersham, RPN2106). Primary antibodies used: anti-αTubulin DM1A (T9026, Sigma-Aldrich, 1:5000), anti-NSD2 29D1 (Ab-75357, Abcam, 1:1000), anti-FKBP12 H-5 (sc-133067, 1:1000), anti-FBXO22 FF-7 (sc-100736, 1:400), anti-XIAP E-2 (sc-55551, 1:200) and anti-GADPH 0411 (sc-47724, 1:5000) all purchased from SantaCruz Biotechnology, anti-cMYC D84C12 (#5605, 1:1000), anti-HA-Tag C29F4 (#3724, 1:1000), anti-BRD4 E2A7X (#13440, 1:2000), anti-CRBN D8H3S (#71810, 1:1000) and anti-V5 D3H8Q (#13202, 1:1000) from Cell Signaling Technology, anti-Flag M2 (F1804, Sigma-Aldrich, 1:1000) and anti-BRD3 (#A302-368A, Bethyl Laboratories, 1:1000). Secondary antibodies used: anti-rabbit IgG, HRP-linked (#7074, 1:10000) and anti-mouse IgG, HRP-linked (#7076, 1:10000) both from Cell Signaling Technology.

## Co-immunoprecipitation of FBXO22-FKBP12, FBXO22-BRD4s or CRBN-BRD4s

HEK293T or HEK393T Nluc-3xFlag-FKBP12 cells were seeded in 10 cm dishes (6 × 10⁶ cells/dish) and incubated overnight at 37 °C to attach. The following day, each 10 cm dish was transiently transfected with 3 μg of the appropriate constructs as specified in the figure legends using PEI for 18 h, before being expanded into 2 × 10 cm dishes. 48 h post-transfection cells were pre-treated with DMSO or carfilzomib for 1 h and then co-treated for 4 h with the appropriate compounds specified in the figure legends. Post-treatment, cells were collected and washed 3x with ice-cold PBS, and lysed in 250 μl of lysis buffer (50 mM Tris-HCl pH 7.4, 150 mM NaCl, 0.1% Triton-X-100, 1 mM EDTA, 5 mM MgCl2, 5% glycerol) freshly supplemented with the 100X Halt™ Protease Inhibitor Cocktail for 20 min on ice. Lysates were cleared at 20.000 rcf spinning down for 20 min and the lysate was normalized with BCA. 200 μg of protein/condition were boiled with 4x LDS at 95 °C for 5 min (Input fraction). In the meantime, 20 μl of Pierce Anti-HA Magnetic Beads (Thermo Fisher Scientific, 88836) per condition were washed with TBS. 1 mg of lysate, adjusted to 200 μL with lysis buffer was incubated with 20 μL beads overnight at 4 °C on rotating wheel. The following day, the beads were separated from the flow-through using a magnetic rack, washed 3x with TBS-T and eluted in 2X Bolt™ LDS Sample Buffer by boiling at 95 °C for 10 min. Western-blot analysis was performed as described above, with 20 μg input and 10% of the IP fraction loaded to 4-12% Bis-Tris gels.

## NanoLuc® Binary Technology (NanoBiT) complementation assay

8 × 10⁵ HEK293T cells/well were seeded in 6-well plates overnight to attach before being transfected with 500 ng of each plasmid SmBiT-FBXO22 and LgBiT-FKBP12, or SmBiT FBXO22 C228A/C326A and LgBiT-FKBP12 using PEI. The cells were incubated overnight and then seeded into 96-well flat, black bottom plates (Costar) at a density of 0.5 × 10⁵/well. The next day, media was removed, cells were gently washed twice with PBS and fresh Opti-MEM I Reduced Serum Medium (Gibco, 31985062) with or without FCS supplemented with 1 μM carfilzomib and 1:20 Vivazine™ (Promega, N2581) was added onto the cells and let to calibrate for 1.5–2 h. Before treating with the desired compounds, baseline luminescence was measured at timepoint 0, using VICTOR™ Multilabel Plate Reader (Perkin Elmer). After treatment, the luminescence was measured with 2 s interval at timepoints indicated in the figure legends.

## Diamine oxidase treatments

Diamine Oxidase from porcine kidney (Sigma-Aldrich, D7876) was prepared fresh in PBS at a concentration of 10 mg ml⁻¹. For the time/dose-dependent SP3CHO quantification experiments in PBS, 10 μM SP3N were incubated with 40 μg diluted DAO in final volume of 100 μL PBS, at 37 °C and 5% $CO_2$ humidified incubator. The metabolites were extracted by adding 200 μL MetOH and subjected to UPLC-MS/MS analysis. For the FACS-based degradation experiments with cells, 10 μM SP3N were treated with DAO as above, for 4 h at 37 °C and 5% $CO_2$ humidified incubator. The DAO-pretreated SP3N solution was added (1:10) on 0.5 × 10⁶ KMB7 iCas9 cells with the FKBP12-BFP-P2A-mCherry reporter that were washed 3x with PBS to remove FCS and resuspended in Opti-MEM -FCS or Opti-MEM supplemented with 10% FCS. Degradation was measured 16 h post-treatment using FACS.

## Design and construction of a ubiquitin-focused sgRNA library

A custom-made focused sgRNA library targeting 1301 ubiquitin-associated human genes with 6 sgRNAs per gene was designed based on the VBC score[60]. Predicted 20mer sgRNA sequences containing a G in the first three 5'-positions were trimmed to the first G at the 5'-end, while others were extended by a 5'-G, resulting in final sgRNA sequences of 18-21 nt in length. The sequences were synthesized as DNA oligo pool (Twist Bioscience) with overhangs and primer binding sites for cloning as previously described[60] and cloned into pLentiV1-PBS69-U6-sgRNA-IT-EF1as-Thy1.1-P2A-Neo. To this end, the DNA oligo pool was amplified using Q5 High-Fidelity DNA Polymerase (New England Biolabs, M0491) in 48 parallel 50 μl PCR reactions, each containing 10 μL 5X Q5 Reaction Buffer, 1 μL dNTP (10 mM each), 2.5 μL forward primer (10 μM), 2.5 μL reverse primer (10 μM), 1 ng oligo pool template, and 0.5 μL Q5 High-Fidelity Polymerase, using the following thermocycler conditions: 98 °C for 30 s; 14 cycles of 98 °C for 10 s, 70 °C for 30 s; final extension at 72 °C for 2 min. The generated amplicons were purified using the QIAquick PCR Purification kit (Qiagen, 28104) according to the manufacturer's recommendations and used in 10 parallel Golden Gate Assembly reactions, each containing 5 ng purified sgRNA amplicon, 200 ng BsmBI (New England Biolabs, R0739) pre-cut and column purified pLentiV1-PBS69-U6-sgRNA-IT-EF1as-Thy1.1-P2A-Neo vector, 1 μL FastDigest Esp3L (Thermo Fisher Scientific, FD0454), 1 μL T7 Ligase (New England Biolabs, M0318), 2 μL FastDigest Buffer (Thermo Fisher Scientific, B64), 1 mM DTT (Roche, 10197777001) and 1 mM ATP (Thermo Fisher Scientific, R0441), all to 20 μL final reaction volume with $H_2O$ and incubated in a thermocycler with the following conditions: 37 °C for 5 min and 23 °C for 5 min for 40 cycles. Pooled ligations were incubated with 2 μL BsmBI and incubated 2 h at 55 °C and subsequently stored at 4 °C. Pooled ligations were purified by Phenol extraction followed by EtOH precipitation and electroporation into MegaX DH10B T1 (Invitrogen, C640003) with a BioRad Pulser II (Bio-Rad) as recommended by the manufacturer. After 1 h recovery at 37 °C, a dilution series of bacteria was plated to ensure a minimum representation of at least 5000 bacterial colonies per sgRNA. The bacteria were grown at 32 °C on LB agar containing 100 μg ml⁻¹ ampicillin for 16 h and the following day, colonies were scraped, recovered in LB broth by shaking at 220 RPM under antibiotic selection at 32 °C for 5 hr. Plasmid DNA was extracted using the NucleoBond Midi prep kit (Macherey-Nagel, REF740410.50).

## FACS-based CRISPR/Cas9 knock-out FKBP12 stability screen

The FACS-based CRISPR/Cas9 FKBP12-BFP stability screen was performed as previously described[31]. Briefly, the lentiviral library containing the UPS-focused sgRNA library (Supplementary Data 2) was generated as described above and used to transduce KBM7 doxycyclin (DOX)-inducibleCas9 (iCas9) FKBP12-BFP-P2A-mCherry cells at a multiplicity of infection (MOI) of 0.1 and 1000x library representation. 10 days post-selection with G418 (1 mg ml⁻¹), 50 × 10⁶ cells were DOX-induced (0.4 μg ml⁻¹, PanReac AppliChem, A2951) for 3 days and

treated with DMSO, 100 nM dFKBP1 or 250 nM SP3N (1000x stocks in DMSO) for 16 h in 2 biological replicates.

Post-treatment, cells were incubated with anti-CD90.1/Thy-1.1-APC (Biolegend, 202526, 1:400), Zombie NIR™ Fixable Viability Dye (BioLegend, 423105, 1:1000) and Human TruStain FcX™ Fc Receptor Blocking Solution (BioLegend, 422301, 1:400), for 10 min at 4 °C, fixed with BD Fixation buffer 4% (Thermo Scientific™ Pierce™, BD 554655) for 45 min at 4 °C, protected from light and stored in PBS + 5% FCS + 1 mM EDTA overnight at 4 ˚C. The next day, cells were sorted on a BD FACSAria™ Fusion (BD Biosciences) using a 100 μm nozzle. The aggregates were excluded based on the forward scatter and side scatter channels and the live cells (Zombie NIR⁻) were gated for Cas9⁺Thy1.1-APC⁺ (Supplementary Fig. 6a). In this population, the fractions FKBP12$^{HIGH}$ (5–8%), FKBP12$^{MID}$ (30-35%) and FKBP12$^{LOW}$ (5-8%) were sorted based on the FKBP12-BFP-mCherry expression levels, at a library representation of at least 1200x.

### Library preparation
Next-generation sequencing (NGS) library preparation of sorted cell fractions was performed as previously described[31,59]. Briefly, the sorted fractions were lysed in lysis buffer (10 mM Tris-HCl, 150 mM NaCl, 10 mM EDTA, 0.1% SDS), supplemented with proteinase K (1:100, New England Biolabs, P8107S) and SDS (1:100) and incubated overnight at 55 °C. After treatment with DNAse-free RNAse digest (Thermo Fisher Scientific, EN0531), the gDNA was extracted using 2 rounds of extraction with phenol (Sigma-Aldrich, P4557) followed by precipitation with isopropanol (Sigma-Aldrich, I9516) overnight at −20 °C.

To generate barcoded NGS libraries a two-step PCR protocol and AmpliTaq Gold DNA polymerase (Thermo Fischer Scientific, 4311806) were used, with the first PCR step to introduce unique barcodes to each sample and the second PCR step to introduce the standard Illumina adapters. Mag-Bind® TotalPure NGS beads (Omega BIO-TEK, M1378) were used to purify the amplified DNA, which was pooled and sequenced on a HiSeq 3500 platform (Illumina).

### Screen analysis
The screen analysis was performed using the crispr-process-nf Nextflow workflow, from https://zenodo.org/records/11445611 and the crispr-mageck-nf Nextflow workflow from https://zenodo.org/records/11445588 as previously described[31]. The median normalized read counts calculated with MAGeCK (0.5.9)[61] were used to compare the FKBP12$^{HIGH}$ or FKBP12$^{LOW}$ versus the respective FKBP12$^{MID}$, per treatment.

### Analysis of *FBXO22* and *CRBN* expression in cancer versus healthy tissues
For the comparison of the expression profiles of *FBXO22* and *CRBN* in healthy and cancer tissues, we extracted mRNA expression data from the Gene Expression Profiling Interactive Analysis, GEPIA2[62], an online platform for integrating RNA sequencing data from The Cancer Genome Atlas (TCGA, https://www.cancer.gov/tcga) and the Genotype-Tissue Expression project of normal tissues (GTEx, https://gtexportal.org/home/).

### Sample preparation and TMT-labeling
Frozen HEK293T pellets were lysed using 8 M urea and 200 mM EPPS at pH 8.5 with protease inhibitors. Samples were then sonicated using a probe sonicator (twenty 0.5-s pulses at level 3). The total amount of protein per sample was determined using a BCA assay. A total of 50 μg of protein was aliquoted for each condition. Protein extracts were reduced using 5-Tris (2-carboxyethyl) phosphine hydrochloride (TCEP) for 15 min at RT. Samples were then treated with 10 mM iodoacetamide for 30 min in the dark at RT followed by precipitation using chloroform/methanol as previously described[63].

After precipitation, samples were digested overnight using LysC and trypsin (1:100 enzyme/protein ratio) at 37 °C using a ThermoMixer set to 1,200 rpm. Following overnight digestion, peptides were labeled with TMTpro 16-plex reagents at a 1:2 ratio by mass (peptides/TMT reagents) for 1 h with constant shaking at 1,200 rpm. Excess TMT reagent was quenched using hydroxylamine (0.3% final concentration) for 15 min at RT. Next, samples were mixed 1:1 across all TMT channels and the pooled sample was fully dried in a Speedvac.

### Basic pH reversed-phase fractionation of TMT-labeled peptides
A 100-mg Sep-Pak solid-phase extraction cartridge was used to desalt the dried, pooled peptide sample, as previously described[63]. The desalted peptide sample was dried in the Speedvac, resuspended (10 mM ammonium bicarbonate, 5% acetonitrile, pH 8.0 buffer) and fractionated into a 96-deep-well plate with basic pH reversed-phase HPLC using an Agilent 300 extend C18 column, and a 50 min linear gradient in 13–43% buffer (10 mM ammonium bicarbonate, 90% acetonitrile, pH 8.0) at a flow rate of 0.250 ml min⁻¹. The fractionated peptide mixture was combined into 24 fractions as previously described, and 12 non-adjacent fractions were desalted using StageTips[63]. 40% of the sample (resuspended in 10 μl of 5% acetonitrile 5% FA) was injected for analysis on an Orbitrap Lumos utilizing a high-resolution MS2-based method.

### Liquid chromatography and tandem mass spectrometry
For the mass spectrometry data collection, Orbitrap Fusion Lumos instruments coupled to a Proxeon NanoLC-1200 UHPLC were used. Peptide separation was achieved with a capillary column (35 cm long, 100 μm diameter) packed with Accucore 150 resin (2.6 μm, 150 Å; ThermoFisher Scientific), at a flow rate of 425 nL min⁻¹. The MS1 spectrum was acquired (Orbitrap analysis, resolution 60,000, 350-1350 Th, automatic gain control (AGC) target 100%, maximum injection time 118 ms), for ~90 min per fraction, followed by the high-resolution MS2 stage consisting of fragmentation by higher energy collisional dissociation (HCD, normalized collision energy 35%) and analysis using the Orbitrap (AGC 200%, maximum injection time 120 ms, isolation window 0.6 Th, resolution 50,000). Data acquisition was performed using the FAIMSpro with the following parameters: dispersion voltage (DV): 5,000 V; compensation voltages (CVs): −40V, −60V, and −80V; TopSpeed parameter: 1 sec per CV.

### Mass spectrometry data analysis
For data searches the open-source Comet algorithm (release_2019010), following a previously described pipeline, and a customized FASTA-formatted database incorporating common contaminants and reversed sequences (Uniprot Human, 2021), were used[64–67]. The ensuing parameters were employed: 50 PPM precursor tolerance, fully tryptic peptides, fragment ion tolerance of 0.02 Da, a static modification by TMTPro16 (+304.2071 Da) on lysine and peptide N-termini, carbamidomethylation of cysteine residues (+57.0214 Da) included as static modification and oxidation of methionine residues (+15.9949 Da) included as a variable modification. To achieve a false discovery rate (FDR) of <1%, the peptide spectral matches were filtered using linear discriminant analysis with a target-decoy strategy. Further filtration ensured a final protein-level FDR of 1% at the dataset level, and proteins were grouped. Reporter ion intensities were adjusted to rectify impurities during the synthesis of different TMT reagents in alignement with the specifications of the manufacturer. Each MS2 spectrum required a total sum signal-to-noise (S/N) of all reporter ions of 160 for quantification. The S/N measurements of peptides corresponding to proteins were summed and normalized to ensure equal loading across all channels. Finally, protein abundance measurements were scaled to achieve a summed signal-to-noise for each protein across all channels equaled 100 (relative abundance measurement).

## In-lysate reactive cysteine profiling

The streamlined reactive cysteine profiling was performed as described in previous work[43,44]. Briefly HEK293T cells were grown in DMEM (Corning) supplemented with 10% FBS and 1% Penicillin/Streptomycin to near confluent, collected, washed twice with cold PBS and were resuspended with lysis buffer (PBS, pH 7.4, 0.1% NP-40) and sequentially homogenized by syringe equipped with 21-gauge needle and probe sonicator (5 min, 3-s on, 5-s off, 50% amp) on ice. Soluble native proteome was collected after centrifugation at 1400 g for 5 min and protein concentration was measured by BCA assay.

To profile reactive cysteine, 15 μL lysate (2 μg/μL), containing 30 μg HEK293T native proteome with spike-in of 0.15 μg recombinant FBXO22-SKP1, was loaded into 96-well plate. 5 μL of compound solution in lysis buffer was added to the plate for final concentrations as described in figure legend and incubated for 1 h at RT. 5 μL of DBIA solution in lysis buffer was added to a concentration of 500 μM and incubated in the dark for 1 h at RT. 3 μL SP3 beads (1:1 mixture of hydrophobic and hydrophilic type, 50 mg ml⁻¹, Cytiva) and 30 μL ~98% ethanol supplemented with 20 mM DTT were added to the plate and incubated for 15 min with mild shaking. Beads were washed once with 200 μL 80% ethanol and resuspended in 25 μL lysis buffer supplemented with 20 mM IAA and incubated in the dark for 30 min with vigorous shaking. 50 μL ~98% ethanol supplemented with 20 mM DTT were added to the mixture followed by beads-based clean-up and 2x washes using 80% ethanol. 30 μL 200 mM EPPS buffer (pH 8.5) containing 0.3 μg Lys-C were added to the remaining beads. After 3 h incubation at RT, 5 μL EPPS buffer containing 0.3 μg trypsin was added and incubated with beads at 37 °C overnight. To the mixture of digested peptides and beads, 9 μL acetonitrile and 6 μL TMT (10 μg/μL) reagent were sequentially added, followed by gentle mixing at RT for 60 min. The reaction was quenched by adding 7 μL 5 % hydroxyl amine and all TMT-labeled samples were combined, dried using a SpeedVac and then desalted using a 100-mg Sep-Pak column.

The desalted TMT-labeled peptides were resuspended in 460 μL of 100 mM HEPES buffer (pH 7.4), 80 μL Pierce™ High Capacity Streptavidin Agarose (Thermo Fisher Scientific, 20359) were added and the mixture was incubated at RT for 3 h. The resulting mixture was then loaded on a Ultrafree-MC centrifugal filter (hydrophilic PTFE, 0.22 μm pore size) and centrifugated at 1000 g for 30 seconds. Beads were washed sequentially with 300 μL 100 mM HEPES (pH 7.4) with 0.05% NP-40 twice, 350 μL 100 mM HEPES (pH 7.4) three times and 400 μL H₂O once. Peptides were eluted sequentially by 1) elution buffer (80% acetonitrile, 0.1% formic acid) with 20-min incubation at RT, 2) elution buffer with 20-min incubation; 3) elution buffer with 10-min incubation at 72 °C. The combined eluent was dried in a SpeedVac and desalted via StageTip prior to LC-FAIMS-MS/MS analysis.

## LC-FAIMS-MS/MS analysis

Enriched cysteines resuspended in 5% ACN and 5% FA were separated on a capillary column (100 μm diameter, packed with 30 cm of Accucore 150 resin), using a 180-min method on a Proxeon NanoLC-1200 UPLC system. Data collection was performed using a high-resolution MS/MS method on an Orbitrap Eclipse mass spectrometer connected to a FAIMS Pro. A 2-shots analysis workflow was followed using a set of three FAIMS compensation voltages (CVs): 1) −60V, −45V and −35V and 2) −70V, −55V and −30V. MS1 scans were collected in the Orbitrap (resolution setting of 60 K, mass range of 400–1600 $m/z$, AGC at 100%, maximum injection time of 50 ms). Data-dependent MS2 scans were acquired in Top Speed mode (cycle time of 1 s, HCD with collision energy of 36) and were collected in the Orbitrap (resolution of 50 K, fixed scan range of 110-2000 $m/z$, and 500% AGC with maximum injection time of 86 ms). A dynamic exclusion of 120 s with a mass tolerance of ±10 p.p.m was chosen. The flowthrough was separated using a 60-min method and analyzed by FAIMS-MS/MS in data-dependent analysis in similar setting as analyzing cysteine samples.

## Data analysis for cysteine identification, localization, and quantification

A workflow similar to the mass spectrometry data analysis above was followed. The raw files were searched using the Comet search engine (ver. 2019.01.5)[68] with the Uniprot human proteome database (downloaded 11/24/2021) with contaminants and reverse decoy sequences appended. The following parameters were used: precursor error tolerance: 50 p.p.m., fragment error tolerance: 0.9 Da, static modifications: Cys carboxyamidomethylation (+57.0215) and TMTpro (+304.2071) on Lys side chains and peptide N-termini, variable modifications: methionine oxidation (+15.9949) and DBIA-modification on cysteine residues (+239.1628). Peptide spectral matches were filtered to a peptide FDR of <1%[64,65], and further filtered to obtain a 1% protein FDR at the entire dataset level[69]. Cysteine-modified peptides were filtered for site localization using the AScorePro algorithm with a cutoff of 13 ($P < 0.05$) as previously described[66,70]. Only unique peptides and cysteine sites were summarized from all PSMs and reported. To quantify TMT reporters in each MS2 spectrum, a total sum signal-to-noise ratio of all reporter ions totaling 180 (for TMTPro 18-plex) was required with <3 missing values. To address variations in loading, quantitative values were normalized to ensure an equal sum of signal for all proteins across each channel.

## Protein expression and purification

All proteins were of human origin. Ubiquitin was expressed tag-less in BL21(DE3) RIL *E. coli* cells. UBA1 was expressed as a GST-TEV fusion in *Spodoptera frugiperda* cells. Following glutathione-affinity purification and TEV protease cleavage the protein was further purified using ion exchange and size exclusion chromatography. UBE2L3, UBE2R2 and ARIH1 were expressed as GST-TEV fusions in BL21(DE3) RIL. Following glutathione-affinity purification and TEV protease cleavage the protein was further purified using ion exchange and size exclusion chromatography. CUL1 and GST-TEV-RBX1 were co-expressed in *Spodoptera frugiperda* cells using two baculoviruses. Following glutathione-affinity purification and TEV protease cleavage the protein was further purified using ion exchange and size exclusion chromatography. CUL1-RBX1 complex was neddylated using previously described procedures[71]. FBXO22 and SKP1 were cloned into a bicistronic pAceBacDual-based vector, as follows. FBXO22 was cloned downstream of Strep-tag II-TEV under control of the polyhedron promoter. SKP1 was placed downstream of the p10 promoter. SKP1-Fbox protein complexes were expressed in *Spodoptera frugiperda* cells. Following Strep-tag II affinity purification, Strep-tag II was removed by treatment with TEV-protease at 4 °C. The protein was further purified by ion exchange and size exclusion chromatography in 25 mM HEPES, 150 mM NaCl, 2 mM DTT, pH 7.5. The C326A mutant of FBXO22 was prepared by Quikchange (Agilent, 200513). FKBP12 was cloned downstream of GST-TEV into pGEX with a Gly-Ser motif at the TEV-cleavage site. The Gly-Ser motif was used as a handle for labeling of FKBP12 with a fluorescent peptide by sortase-mediated transpeptidation[72]. The protein was expressed in BL21 (DE3) RIL *E. coli*. Following glutathione affinity purification and TEV-mediated cleavage, FKBP12 was further purified using size exclusion chromatography.

## Peptides

Peptide was prepared by the Max Planck Institute of Biochemistry Core Facility (>90% purity) and used as received. The peptide to fluorescently label FKBP12 had the following sequence: Carboxyfluorescein-GSGGLPETGG.

## Fluorescent labeling of FKBP12

FKBP12 (100 μM) was mixed with fluorescent peptide (200 μM) in 50 mM Tris, 150 mM NaCl, 1 mM TCEP, 10 mM Ca(OAc)₂, pH 7.5. SrtA 4 M (10 μM) was added and the reaction was incubated for 5 h at 4 °C. The reaction product was purified by size exclusion chromatography.

## Multiturnover ubiquitylation assay

Fluorescent-FKBP12 (0.5 μM), CUL1–NEDD8-RBX1 (0.5 μM), FBXO22-SKP1 (0.5 μM), ARIH1 (0.4 μM), UBE2L3 (2.0 μM) and UBE2R2 (2.0 μM) were mixed with buffer (50 mM HEPES, 100 mM NaCl, 7.5 mM MgCl$_2$, 5 mM ATP, 0.5 mg ml$^{-1}$ BSA, pH 7.5) and either DMSO or compounds (10 μM). The reaction was initiated by the addition of UBA1 (0.1 μM). The reaction was allowed to proceed at RT and time points were removed at 0 and 30 min, quenched by mixing with 3x SDS-PAGE buffer (150 mM Tris-HCl, 20 vol% glycerol, 30 mM EDTA, 4% SDS and 4 vol% b-mercaptoethanol). Time points were resolved on hand-cast 4-22% SDS-PAGE gels. Gels were visualized on a Typhoon 9410 Imager (cytiva).

## Pulse-chase autoubiquitylation of FBXO22

UBE2L3 (10 μM) was charged with fluorescent-ubiquitin (15 μM) by UBA1 (0.4 μM) in 25 mM HEPES, 100 mM NaCl, pH 7.5 in the presence of MgCl$_2$ (5 mM) and ATP (1 mM) at RT. The charging reaction was allowed to proceed for 15 min before quenching with apryase (8 U ml$^{-1}$) for 5 min at 4 °C. UBE2L3 ~ UB* (0.4 μM) was added to ARIH1 (0.3 μM), CRL1–NEDD8 (0.5 μM), SKP1-FBXO22 (0.5 μM). Time points were removed at indicated times and resolved by SDS-PAGE analysis and in-gel fluorescence analysis.

## Intact Mass determination of FBXO22-compound complex

FBXO22-SKP1 variants (20 μM) were mixed with buffer (25 mM HEPES, 100 mM NaCl, pH 7.5). 100 μM of the respective compound were added and the mixture was incubated for 10 min on ice prior to ESI-LCMS analysis. LCMS analysis for intact mass determination was carried out on a microTOF Bruker Daltonik instrument equipped with an Agilent 1100 HPLC system. Samples were resolved on a Phenomenex AerisTM 3.6 μm WIDEPORE C4 100 mm×2.1 mm ID, 200 Å pore size column with eluents H$_2$O + 0.05 vol% TFA (Buffer A) and MeCN + 0.05 vol% TFA (Buffer B). Elution was achieved with a Buffer B gradient of 20–95% over 16 min at a flow rate of 250 μL/min. MS mode was set to positive detection and a mass range of 800–3000 *m/z*. Raw MS data was analyzed in CompassTM Data Analysis software from Bruker Daltonik and deconvoluted with "Maximum Entropy".

## nanoDSF

SKP1-FBXO22, wt and C326A, were diluted with 25 mM HEPES, 150 mM NaCl, 1 mM TCEP, pH 7.5 to 2.7 μM. Protein heat denaturation was measured using nanoDSF (Prometheus, Nanotemper). 10 μL of protein solution was loaded in each capillary. Tryptophan fluorescence was measured at 330 and 350 nm with 40% gain. Following temperature stabilization at 20 °C for 20 min, heat denaturation was measured at a rate of 1 °C/min. Measurements were performed in duplicates.

## Ultra-performance liquid chromatography-mass spectrometry (UPLC-MS/MS) for SP3N and SP3CHO

80% methanol cell extracts (or solutions without cells) were evaporated to dryness using a soft nitrogen flow. Samples were reconstituted with 50 μL of HPLC-grade methanol and vortexed. Samples were analyzed by UPLC-MS/MS using a Waters Acquity UHPLC system coupled to a Waters Xevo TQMS triple quadrupole mass spectrometer. The conditions for the equipment were optimized before the sample analysis to obtain the best selectivity and sensitivity. The analytes were separated using reverse phase liquid chromatography on an Agilent ZORBAX Eclipse Plus C18 Rapid Resolution HD (1.8 μm, 2.1 × 50 mm) analytical column equipped with a pre-column, running a gradient with solvent A (water with 0.1% formic acid) and solvent B (acetonitrile with 0.1% formic acid), and keeping the column compartment at 40 °C. The mass spectrometer was run in positive electrospray ionization (ESI) mode, monitoring 2 MRM transitions per compound: for SP3CHO *m/z* 727.404 > 472.26 and 727.404 > 99.02: for SP3N 728.628 > 473.409 and 728.628 > 270.223; and for SP3NAc 770.638 > 515.436 and 770.638 > 86.043. For the absolute quantification of the compounds, a 10-point external calibration curve was recorded using neat standards, from 0.04 to 10,000 nM. Excellent linearity was obtained in all the range ($R^2 > 0.996$). Quantification was executed using the software TargetLynx XS V4 S2 SCN986.

## Reporting summary

Further information on research design is available in the Nature Portfolio Reporting Summary linked to this article.

## Data availability

The mass spectrometry data for the global proteomics generated in this study have been deposited in the ProteomeXchange Consortium via the PRIDE partner repository with the accession identifier PXD049330[73]. The mass spectrometry data for the TMT-ABPP experiment have been deposited in the ProteomeXchange Consortium with the accession identifier PXD051803. Source data of all graphs and uncropped gels and blots are provided in the "Source Data" file. Source data for Figs. 1e, 2a, Supplementary Fig. 2a and Supplementary Fig. 4g, h are included in Supplementary Data 1–4. The gating strategies applied for FACS analyses and cell sorting are provided in Supplementary Fig. 6. Source data are provided with this paper.

## Code availability

The code for the analysis of the FACS-based CRISPR-KO screens is available on GitHub [https://zenodo.org/records/11445588, https://zenodo.org/records/11445611].

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

## Acknowledgements

CeMM and the Winter lab are supported by the Austrian Academy of Sciences. The Winter lab is further supported by funding from the European Research Council (ERC) under the European Union's Horizon 2020 research and innovation program (grant agreement 851478), as well as by funding from the Austrian Science Fund (FWF, projects P7909, P36746 and P5918723) and an Aspire Award from the Mark Foundation for Cancer Research. This work in the Schulman lab was supported by the Max Planck Society, and the European Research Council (ERC) under the European Union's Horizon 2020 research and innovation program (grant agreement No 789016-NEDD8Activate). J.F. is supported by a postdoctoral fellowship from the Peter und Traudl Engelhorn-Stiftung. We thank the Core Facility Flow Cytometry of the Medical University of Vienna for access to flow cytometry instruments and assistance with flow cytometric cell sorting as well as the CeMM Biomedical Sequencing Facility for NGS sample processing, sequencing, and data curation. We are grateful to all the members of the Winter lab for helpful discussions. We thank Stephan Uebel and the Biochemistry Core Facility of Max Planck Institute of Biochemistry for peptide synthesis and Maria Victoria Sanchez Caballero, Barbara Steigenberger and the Mass Spectrometry Core Facility of Max Planck Institute of Biochemistry for intact mass analysis. We thank Susanne von Gronau for protein expression in insect cells.

## Author contributions

C.K. designed and executed most of the described experiments, analyzed data, wrote the manuscript, and made figures. J.A.C. synthesized compounds. J.F. performed in vitro ubiquitylation assays and intact MS. J.L. produced recombinant neddylated SCF$^{FBXO22}$ and FKBP12 and performed preliminary in vitro biochemical experiments. F.O. developed the NanoBiT assay. K.D. and J.A.P. performed and analyzed the global proteome experiment. K.Y. performed and analyzed the Cys-ABPP proteome-wide profiling. G.T. synthesized compounds. C.S.H. performed mutagenesis studies. M.H. developed the FKBP12 stability reporter. N.S.S. supported CRISPR screens. J.S.A. performed metabolomics experiments. M.F. cloned UPS-focused sgRNA library. F.A. annotated UPS-focused sgRNA library. J.T.H. supervised and analyzed metabolomics experiments. J.Z. supervised UPS-focused sgRNA library design. S.K. supervised metabolomics. S.P.G. supervised proteomics experiments. B.A.S. supervised in vitro experiments. G.E.W. conceptualized the study, wrote the manuscript, and has overall responsibility for the presented study.

## Competing interests

S.K. and G.E.W. are scientific founders and shareholders of Proxygen and Solgate. G.E.W. is on the Scientific Advisory Board of Nexo Therapeutics. The Winter lab received research funding from Pfizer. B.A.S. is a member of the scientific advisory boards of Proxygen and BioTheryX, and a co-inventor of intellectual property licensed to Cinsano. The remaining authors declare no competing interests.
