## [Peer Review File · Nature Communications]

Alkylamine-tethered molecules recruit FBXO22 for targeted protein degradationREVIEWER COMMENTS

Reviewer #1 (Remarks to the Author):

Summary:

Kagiou et al. clearly and concisely describe the mechanism of action of a small-molecule degrader SP3N of FKBP12. Through chemical biology experiments, they describe how SP3N degrades FKBP12 in a UPS- and CRL-dependent fashion. This is critical because there is no obvious E3-targeting moiety on SP3N to implicate it via this mechanism.

The authors proceed to identify the E3 involved in the process by a Ubiquitin-proteasome-system-focused CRISPR/cas9 knock-out screen. Here, they identify FBXO22 as the E3 ligase primarily responsible for FKBP12 degradation by SP3N. They confirm this interaction by targeted experiments, including genetic reconstitution of degradation in FBXO22-KO cells, co-IP of FKBP12 and FBXO22 in the presence of SP3N, and nanoluciferase complementation via induced proximity of FKBP12 and FBXO22 in the presence of SP3N. While the previous experiments were performed in cells, the authors fully reconstitute the neddylated SCFFBXO22 complex using purified proteins and demonstrate ubiquitylation of FKBP12 in the presence of SP3N. Importantly, they identify one cysteine residue that is crucial for this interaction and repeat the aforementioned experiments with the cysteine-to-alanine change in FBXO22 to validate their findings.

To translate these findings broadly, the authors perform a series of experiments to identify that the active form of SP3N includes an aldehyde motif. This chemical alteration is shown to be performed in a cellular media, not within cells. Finally, the authors show that literature degraders of XIAP and NSD2 also most likely work by hijacking FBXO22 via a similar mechanism to SP3N. However, this was not generalizable to JQ1-alkylamine degraders designed to target BRD4, which poses a puzzle for further development.

Overall, the paper is well-written and presented, and will be of interest to the TPD/E3/UPS community. Although the community is not lacking for FKBP12 degraders, and it is unclear why the authors are pursuing that in the first place, the serendipitous discovery of FBXO22-based degradation is noteworthy. A pre-print describing a similar mechanism of action with an NSD2 degrader makes this a timely addition, since Kagiou et al. utilize orthogonal techniques and end up in the same place.

Overall, the article is recommended for publication, after addressing the following

comments:

Major points:

1. AlphaFold has a reliable prediction for the structure of this protein. As such I don't agree with the literature definitions of "N- and C-terminal FIST domains" and conclusions based on such. It appears as if the C-terminal domain of FBXO22 can be split into three "blades" - [65-181], [183-283], and [284-403], none of which match up to the literature cited. While it is outside the scope of this paper to support that claim with a structure, I think it would be more accurate, citing AlphaFold, to reword or eliminate the references to the N-FIST and C-FIST. It would be interesting to predict the location of C326 and its accessibility by SP3N. Is it solvent accessible or buried in protein core? Any evidence that C326A does not cause misfolding of the protein leading to loss of function?
2. It is important to show a gel and SEC profile of purified FBXO22, and the mutant, since it has not been reported previously. Related, for figures 3H and 4D please correct the title to say that it is the FBXO22:Skp1 complex.
3. Additionally, the article would benefit from a graphical / cartoon / structural model of the hypothesized interaction.
4. In order to designate SP3N a prodrug, the conversion to parent drug must be observed in vivo (in animals). No in vivo experiments were reported in this paper. Without in vivo confirmation, the definition of a prodrug has not been met and should be used loosely. Minimally, a PK study should be included to support designation as prodrug? This seems critically important given the in vitro quantification results in Figure 3d.

Minor points

1. Why proteomics at 16 hr when the timecourse shows 8hr is good enough and author suggests covalent modification plateaus in 2 hr? Across the study, treatment times vary – is there a reason?
2. Pg 7, lines 46-50, the authors make the SP2-chloroacetamide and acrylamide molecules. However, the bulk of the main text has been focused on the 3-PEG (vs 2-PEG)-containing molecules. Why did the authors go this route?
3. For all the reported substrates of FBXO22, is there a consensus degron? It seems to be a "good degrader" E3; is there anything known about natural substrates using the same PPI interface exploited SP3N?

4. What is in the UPS-focused CRISPR/Cas9 library? A focused sgRNA library targeting 1301 ubiquitin-associated human genes with 6 sgRNAs per gene was designed – is it commercial or custom built?
5. There are errors in captions 3i) and 4e) for the expected mass of the FBXO22-SP3CHO adduct – please correct.
6. Page 9, line 20 says “all three” assayed degraders, but you reference more than three in the first sentence. Please make it clear that “all three” does not refer to the BRD4-targeting compounds (since it did not work).
7. Figure 2D shows proof of recruitment of FBXO22 for degradation by coimmunoprecipitation. The FKBP12 is a double band in the IP:HA and a single band in the Input. Only the bottom band shows rescue from SP3N degradation by carfilzomib. Can you explain the two bands and what the upper band represents? Non-specific antibody result? Degradation product?
8. Figure 3, the work regarding quantification around conversion of SP3N to SP3CHO with and without cells (Figure 3b, c, d, e). I would encourage the authors to edit the figure to data critical to support quantification levels. The figure needs to be understandable without significant deep-dive into the text – simplify.

Reviewer #2 (Remarks to the Author):

In this paper, the authors describe the identification of compound SP3N (SLF-PEG3-NH₂) that is capable of recruiting the SCF-FBXO22 ligase to induce the polyubiquitylation of FKBP12, leading to its degradation by the proteasome, and further unravel the mechanism of action of SP3N. They nicely dissect the serendipitous discovery that SP3N functions as a prodrug and tie their results back to previously disclosed compounds (UNC 5153 & GNE compound 10). The authors' findings show a higher potency of degradation compared to previous studies on covalent degraders, which strengthens the case for further exploration of FBXO22 as a target for drug discovery. I anticipate that the TPD community (and scientific community at large) will be very enthusiastic about this study. Overall, after appropriate revisions, I believe that the paper would be a suitable fit for publication in Nature Communications.

Minor comments:

1. Please add statistical analysis to all the main and supplementary figures.

2. Discussion section

a. Please discuss in more details if there are any potential advantages (such as tissue expression and selectivity, disease specificity, unique binding sites etc.) of harnessing FBXO22 compared with the currently available E3 ligases. For example, it would be nice to include a supplemental figure that displays the expression profile of FBXO22 in normal and disease (e.g. GTEx and TCGA) compared to CRBN.

b. The authors comment that native substrates have been mapped for FBXO22. Do the authors anticipate if conjugation to C326 would disrupt native substrate binding? If so, are there any toxicity = implications?

Major comments:

1. In figure 3D, 3F (+FCS group experiments), the SP3CHO outperforms SP3N in the presence of FCS. Since it is stated by authors that FCS is not changing SP3CHO chemical properties, what is the reason for reduced activity? (permeability?)

2. In figure 3F, there is a significant hook effect observed at the highest tested dose for SP3CHO in -FCS group, however, this is not the case for +FCS group. i) Please explain why there is such a big hook in the absence of FCS for SP3CHO? Also, SP3CHO is a bit more potent in the absence of FCS, ii) Is this difference statistically significant? If yes, it is important to show how FCS affects the potency of the compound. For instance, does FCS change cellular permeability, cellular uptake, binding affinity, dissociation kinetics?

3. Based on supplementary Fig. 3e. (FKBP degradation), the authors argue that aldehydes are favored over other electrophiles. This statement seems overly broad with little supporting evidence. For example, the preprint from Xiaoyu Zhang used a chloroacetamide-based degrader and observed comparable degradation potencies. It is likely that differences in warhead or linker are a driving factor regulating potency. To make a stronger case, the authors should (at the least) compare the percent FBXO22 adduct formation between the

compounds listed in Fig. S3e to ensure that differences aren't simply due to reactivity.

4. The authors should run proteome-wide APBB-MA experiment with their degraders. This information would provide a deeper understanding of compound selectivity and general activity of FBXO22 (e.g. percent FBXO22 engagement necessary for TPD).

5. As stated by the authors, there are limits to the generalizability of the approach. However, since proteins are not equally amenable to E3-based degraders and as BRD4 is the only protein tested for this system in this paper, i) It would be nice to test other systems where many targets can be targeted such as using pan-kinase inhibitors as warhead domains. ii) It is also possible that the compound forms a ternary complex (TCF) with BRD4 and FBXO22 but unable to form a favorable TCF and/or access available lysine for ubiquitination. So, it is crucial for this study to at least show whether the generated BRD4 series can bind to FBXO22 (Binary complex formation), and if ternary complex can be formed. iii) Based on the data provided in supplementary Fig. 4b, authors tested BRD4 degradation using BRD4(short)-BFP-P2A-mCherry cells. Is this possible that fusing BRD4 to BFA restricting favorable ternary complex formation and lysine accessibility for E3 ligase FBXO22 and the ubiquitination machinery? Please assess the degradation potency of the BRD4 library against endogenous BRD4 using other systems such as HiBiT tagging systems, or Lumit or immunoblots?

6. In the discussion, the authors make an interesting argument that the prodrug strategy could be exploited to drive tumor restricted degradation by targeting indications where amine oxidases are upregulated. Have the authors tested cell lines that express high vs low to see if selective degradation is observed? This data would strengthen the paper.

7. The mutagenesis experiment showing that C326 is the target cysteine for their degraders is compelling. To provide even stronger evidence, the authors should consider showing that this mutant still engages Skp1 in cells. This would rule out artifacts due to changes in protein folding.

Reviewer #3 (Remarks to the Author):

In the manuscript 'Alkylamine-tethered molecules recruit FBXO22 for targeted protein degradation'

Winter and colleagues identify and rigorously characterise a novel degrader prodrug SP3N that consists of a known binder of FKBP12 appended with a flexibly alkylamine tail. In cells the molecule is metabolized to an active aldehyde that recruits the SCF/FBXO22 ligase through covalent adduction of Cys326 of FBXO22. The authors further demonstrate that their results are in line with a similarly detailed recent study by the Arrowsmith lab that described the same mechanism when investigating alkyl amine-based degraders of NSD2 and XIAP. Whilst the data on the FKBP12 degrader suggest broader applicability of the approach absence of degradation of alkyl amine tethered analogs of the BET inhibitor JQ1 suggest that there are limits to the generalisability of the described concept.

The study is comprehensive, data quality is very good, claims are generally well grounded on data and the observation that alkylamine thethers provide prodrugs leading to reversible covalent linkage to FBXO22 and proximity induced degradation of target proteins is highly interesting and relevant.

My only concern is that this study closely follows work published as a preprint by the Arrowsmith lab and the incremental novelty is very limited.

REVIEWER COMMENTS

Reviewer #1 (Remarks to the Author):

Summary:

Kagiou et al. clearly and concisely describe the mechanism of action of a small-molecule degrader SP3N of FKBP12. Through chemical biology experiments, they describe how SP3N degrades FKBP12 in a UPS- and CRL-dependent fashion. This is critical because there is no obvious E3-targeting moiety on SP3N to implicate it via this mechanism.

The authors proceed to identify the E3 involved in the process by a Ubiquitin-proteasome-system-focused CRISPR/cas9 knock-out screen. Here, they identify FBXO22 as the E3 ligase primarily responsible for FKBP12 degradation by SP3N. They confirm this interaction by targeted experiments, including genetic reconstitution of degradation in FBXO22-KO cells, co-IP of FKBP12 and FBXO22 in the presence of SP3N, and nanoluciferase complementation via induced proximity of FKBP12 and FBXO22 in the presence of SP3N. While the previous experiments were performed in cells, the authors fully reconstitute the neddylated SCFFBXO22 complex using purified proteins and demonstrate ubiquitylation of FKBP12 in the presence of SP3N. Importantly, they identify one cysteine residue that is crucial for this interaction and repeat the aforementioned experiments with the cysteine-to-alanine change in FBXO22 to validate their findings.

To translate these findings broadly, the authors perform a series of experiments to identify that the active form of SP3N includes an aldehyde motif. This chemical alteration is shown to be performed in a cellular media, not within cells. Finally, the authors show that literature degraders of XIAP and NSD2 also most likely work by hijacking FBXO22 via a similar mechanism to SP3N. However, this was not generalizable to JQ1-alkylamine degraders designed to target BRD4, which poses a puzzle for further development.

Overall, the paper is well-written and presented, and will be of interest to the TPD/E3/UPS community. Although the community is not lacking for FKBP12 degraders, and it is unclear why the authors are pursuing that in the first place, the serendipitous discovery of FBXO22-based degradation is noteworthy. A pre-print describing a similar mechanism of action with an NSD2 degrader makes this a timely addition, since Kagiou et al. utilize orthogonal techniques and end up in the same place.

Overall, the article is recommended for publication, after addressing the following comments:

We want to thank reviewer #1 for their kind remarks, insightful suggestions, and positive feedback.

Major points:

1. AlphaFold has a reliable prediction for the structure of this protein. As such I don't agree with the literature definitions of "N- and C-terminal FIST domains" and conclusions based on such. It appears as if the C-terminal domain of FBXO22 can be split into three "blades" - [65-181], [183-283], and [284-403], none of which match up to the literature cited. While it is outside the scope of this paper to support that claim with a structure, I think it would be more accurate, citing AlphaFold, to reword or eliminate the references to the N-FIST and C-FIST. It would be interesting to predict the location of C326 and its accessibility by SP3N. Is it solvent accessible or buried in protein core? Any evidence that C326A does not cause misfolding of the protein leading to loss of function?

We thank reviewer #1 for their detailed comment regarding the terminology used to describe FBXO22 domains. We appreciate their insight into the potential discrepancy between the literature definitions of "N- and C-terminal FIST domains" and the structure provided by AlphaFold predictions. We carefully reviewed the AlphaFold predictions for FBXO22 and agree with the reviewer's assessment. Hence, we eliminated the outdated terms N-FIST and C-FIST domains of FBXO22 to better align with the structural information provided by AlphaFold.

Changes to the manuscript:

- **Page 9, lines 7-9 (changes highlighted):** "To map the functionally required and covalently engaged cysteine residue in FBXO22, we mutated five cysteine residues in the C-terminal

region of FBXO22 (amino acids 143-365) that has been reported to play a role in substrate binding.”

- **Page 12, lines 3-5 (changes highlighted):** “This is of particular interest in light of recent findings by Basu et al., highlighting that additional cysteine residues (C227, C228) of FBXO22 are in principle ligandable with chloroacetamide-based compounds.”
- The term “FIST domain” was removed throughout the manuscript.

Additionally, we appreciate the suggestion to predict the location/accessibility of C326. In the revised Supplementary Fig. 4a we highlighted C326 in the predicted AlphaFold structure of FBXO22. Indeed, based on this prediction, C326 appears to be solvent-accessible and hence accessible for chemical modulation.

Changes to the manuscript:

- A revised Supplementary Fig. 4a panel has been added (see supplementary information):

a

Supplementary Fig. 4a AlphaFold prediction of FBXO22 structure with C326 highlighted in blue.

- The relevant sentences in the manuscript describing this change now read (changes highlighted):

Page 9, line 12-13: “SP3N- or SP3CHO-induced degradation was maintained by all FBXO22 mutations with the exception of C326A (Fig. 4a, b, Supplementary Fig. 4a, b).”

Page 11, lines 51-52: “Our data, alongside the data reported in Nie et al., clearly highlight an essential functional role of the solvent-accessible Cys326 in the C-terminal domain of FBXO22.”

Finally, we appreciate the feedback regarding the potential C326A mutation's impact on FBXO22 folding and activity. Given that this point was also independently raised by reviewer #2, we put emphasis on addressing it with additional recombinant experiments and cellular assays. To this goal, we performed nano differential scanning fluorimetry (NanoDSF) to test the protein stability of the SKP1-FBXO22-WT and SKP1-FBXO22-C326A complexes. These experiments revealed thermal stability of both variants, suggesting that C326A does not affect protein folding (Supplementary Fig. 4c).

Changes to the manuscript:

- The revised Supplementary Fig. 4c panel has been added (see supplementary information):

Supplementary Fig. 4c Normalized fluorescence data from nano differential scanning fluorimetry (NanoDSF) of SKP1-FBXO22-WT and SKP1-FBXO22-C326A. Heat denaturation was measured at a rate of 1 °C/min. Measurements were performed in duplicates.

In addition, we added experimental evidence that the FBXO22-C326A mutant incorporates into a functional CRL complex. CRL ligases are actively disassembled in cells via a process that is dependent on cullin deneddylation via the COP9 signalosome (CSN). Pharmacologic CSN inhibition thus arrests a CRL in a constitutive active state that frequently leads to auto-degradation (particularly on the substrate receptor). Hence, induction of protein degradation of a substrate receptor upon CSN inhibition strongly implies that it is incorporated in an active CRL (Schlierf et al, *Nat Commun*, 2016; Mayor-Ruiz et. al, *Mol Cell*, 2019; Henneberg et. al., *Nat Chem Biol*, 2023). Supporting the observation that the C326A mutation does not affect its functionality, we observed that cellular treatment with the CSN inhibitor CSN5i-03 led to similar degradation profiles of both FBXO22 variants (Supplementary Fig. 4d).

Changes to the manuscript:

- The revised Supplementary Fig. 4d panel has been added (see supplementary information):

Supplementary Fig. 4d Immunoblot of 2HA-FBXO22-WT or 2HA-FBXO22-C326A in HEK293T-FKBP12-BFP-P2A-mCherry FBXO22 KO single clone transduced with 2HA-FBXO22-WT or -C326A cDNAs. Cells were treated with DMSO or 500 nM CSN5i-03 for 1-16 h. α Tubulin is the loading control. Representative plot of $n=2$ independent experiments.

Finally, these findings were further supported by an *in vitro* autoubiquitylation assay which revealed comparable rates of autoubiquitylation of FBXO22-WT and FBXO22-C326A variants, thus again suggesting intact SCF complex activity (Supplementary Fig. 4e).

Changes to the manuscript:

- The revised Supplementary Fig. 4e panel has been added (see supplementary information):

Supplementary Fig. 4e Pulse-chase *in vitro* autoubiquitylation assay of FBXO22 with SKP1-FBXO22-WT and SKP1-FBXO22-C326A. UB*: fluorescent ubiquitin. Representative blot from $n=2$ independent experiments.”

- The relevant paragraph in the manuscript describing the results for all Supplementary Fig. 4c, d, e now reads (page 9, lines 13-20):

“To rule out potential deleterious effects of the C326A mutation on FBXO22, we turned to Nano differential scanning fluorimetry (NanoDSF) which revealed thermal stability (Supplementary Fig. 4c). In addition, FBXO22 WT and C326A are similarly sensitive to pharmacologically induced auto-degradation via COP9 signalosome inhibition (Supplementary Fig. 4d), supporting the notion that the mutant incorporates into a functional SCF complex also in cells.^{40–42} Further, an *in vitro* autoubiquitylation assay demonstrated comparable autoubiquitylation of recombinant FBXO22-WT and FBXO22-C326A variants, hence again suggesting intact SCF complex activity (Supplementary Fig. 4e).”

- The **Methods** section of the manuscript has been updated to include the aforementioned experiments.

2. It is important to show a gel and SEC profile of purified FBXO22, and the mutant, since it has not been reported previously. Related, for figures 3H and 4D please correct the title to say that it is the FBXO22:Skp1 complex.

We thank reviewer #1 for bringing these important points to our attention. We agree that a comprehensive characterization of the purified proteins is essential, thus we included the SEC profiles of the purified FBXO22 WT and C326A mutant in the revised Supplementary Fig. 3e and Supplementary Fig. 4b.

Changes to the manuscript:

- Modified versions of supplementary Fig. 3 (FBXO22-WT SEC profile) and supplementary Fig. 4 (comparison of FBXO22-WT and FBXO22-C326A SEC profiles) have been added (see supplementary information):

e

Supplementary Fig. 3e Size exclusion chromatography (SEC) profile of the recombinant SKP1-FBXO22-WT. Relative absorbance over time is shown.

b

Supplementary Fig. 4b Comparison of SEC profiles of SKP1-FBXO22-WT and SKP1-FBXO22-C326A. Relative absorbance over time is shown.

We additionally provide here a gel of the purified recombinant complexes (Reviewer-only Fig. 1).

Reviewer-only Fig. 1. Gel of the purified SKP1-FBXO22-WT and SKP1-FBXO22-C326A stained with Coomassie Blue.

Finally, we have corrected the titles to depict the SKP1-FBXO22 complex in the revised Fig. 3g (previously annotated as 3h) and 4d.

Changes to the manuscript:

- Figures 3g and 4d: the title FBXO22 was modified to SKP1-FBXO22 (pages 8 & 10, respectively)

3. Additionally, the article would benefit from a graphical / cartoon / structural model of the hypothesized interaction.

We agree that a detailed understanding of the induced interaction and underpinning molecular recognition would be helpful. We in fact have invested a lot of effort into generating a ternary structure via cryo-EM, but failed, until now, to generate a map with sufficient resolution. We hope that by now clearly highlighting the adducted C326 in the AlphaFold model (see Supplementary Fig. 4a), we have sufficiently indicated the interaction site on FBXO22. We believe that going into more depth with protein-protein interaction modeling would be outside the scope of this study and we would prefer to follow up on questions of detailed molecular recognition in a separate study.

4. In order to designate SP3N a prodrug, the conversion to parent drug must be observed in vivo (in animals). No in vivo experiments were reported in this paper. Without in vivo confirmation, the definition of a prodrug has not been met and should be used loosely. Minimally, a PK study should be included to support designation as prodrug? This seems critically important given the in vitro quantification results in Figure 3d.

We thank reviewer #1 for their feedback regarding the definition criteria of the “prodrug” terminology that we did not consider in depth when writing the manuscript. We agree with the concern and understand that this is an important aspect. However, the focus of our study is understanding the molecular mode of action of SP3N and we hence consider this investigation outside the scope of our study. Since running additional pharmacokinetic studies is a significant undertaking requiring additional resources that we don’t have in place (mouse protocols including ethics approval), we would moreover not be able to comply with the timeframe of our revision plan.

Nevertheless, we recognize that reviewer #1 is raising a very valid point here. We have hence decided to address the concerns by changes in the text, now referring to SP3N as a “precursor” instead of a “prodrug” throughout the manuscript. This change aligns with the terminology used in the field and will enhance the clarity and accuracy of our work.

Changes to the manuscript:

- The term “*prodrug*” was eliminated and exchanged to “*precursor*” throughout the manuscript.

Minor points

1. Why proteomics at 16 hr when the timecourse shows 8hr is good enough and author suggests covalent modification plateaus in 2 hr? Across the study, treatment times vary – is there a reason?

We are happy to elaborate on this point. In our study, we employed a variety of methodologies to investigate the mode of action of SP3N. Each chosen timepoint was selected to align with the specific objectives of the assay and corresponding hypotheses. For our proteomics analysis, we opted for a 16-hour timepoint to capture potential later degradation events comprehensively since, in our experience, off-target degradation events can sometimes happen with a delay. In the case of CRISPR sorting to distinguish between FKBP12-BFP^{high} and FKBP12-BFP^{low} populations, we carefully selected the optimal timepoint where degradation enabled clear sorting of these populations. This strategic decision ensured the effectiveness of our sorting process. In our immunoprecipitation and NanoBiT assays, where the focus was on capturing complex formation preceding degradation, we conducted time-courses ranging from minutes up to 4 hours. This approach enabled us to capture the dynamic interactions between proteins and provided insights into the early stages of complex formation. Lastly, based on initial time and dose response experiments indicating increased potency of SP3CHO, we chose an 8-hour timepoint for our assays to maximize degradation efficiency.

2. Pg 7, lines 46-50, the authors make the SP2-chloroacetamide and acrylamide molecules. However, the bulk of the main text has been focused on the 3-PEG (vs 2-PEG)-containing molecules. Why did the authors go this route?

We appreciate reviewer's #1 feedback, and we understand their concern regarding the chosen linker length. We hypothesized that the positioning of the electrophilic warhead could influence the access to the FBXO22 C326. We noticed that the aldehyde present in SP3CHO would likely occupy a similar spatial position as the acrylamide or chloroacetamide of SP2. Therefore, we opted for the PEG2 scaffold due to the alignment with the relative positioning of these electrophilic warheads. Nevertheless, to address potential concerns regarding the activity of the PEG3-based covalent molecules we additionally synthesized SP3-Acry and SP3-Cl. Leveraging our FACS-based assays, we could show that also these analogs are inactive in inducing FKBP12 degradation.

Changes to the manuscript:

- A revised Supplementary Fig. 3g panel has been added (see supplementary information, changes highlighted):

Supplementary Fig. 3g Flow-cytometry based degradation assay in KBM7 iCas9 with the FKBP12-BFP-P2A-mCherry reporter treated with DMSO or 10 μ M SP2N, SP2-acrylamide (SP2-Acry), SP2-chloroacetamide (SP2-Cl), SP3N, SP3-Acry or SP3-Cl for 16 h. For all the flow-cytometry based degradation assays (b, g), the BFP/mCherry ratio was normalized to DMSO and the data is the mean \pm s.d. from $n = 3$ biological replicates.

- The relevant paragraph in the manuscript describing these results now reads (page 8, lines 2-9, changes highlighted):

“Having established that SP3N covalently adducts FBXO22, and based on another recent report that established SCF^{FBXO22} as a ligase that can be harnessed with a covalent, chloroacetamide containing PROTAC, we wanted to investigate if we could replace the aldehyde with alternative warheads and synthesized four additional electrophilic compounds, namely the SP3-chloroacetamide, the SP3-acrylamide and the respective SP2-based analogs.³⁷ Interestingly, none of these compounds exhibited robust FKBP12 degradation, indicating that these SLF-based aldehydes are favored over the other SLF-based electrophiles in inducing FBXO22-dependent protein degradation (Supplementary Fig. 3g).

3. For all the reported substrates of FBXO22, is there a consensus degron? It seems to be a “good degrader” E3; is there anything known about natural substrates using the same PPI interface exploited SP3N?

We appreciate the insightful question regarding the consensus degron for the reported substrates of FBXO22. Although it has been reported that FBXO22 recognizes substrates based on phosphodegrons

(Liu et al., Cell Death Differ, 2022) the precise degron sequence remains to be fully elucidated. Whether endogenous substrates utilize a similar binding interface is an intriguing question but, in our opinion, experimentally outside the scope of this manuscript, but will be something we are going to explore in future studies.

4. What is in the UPS-focused CRISPR/Cas9 library? A focused sgRNA library targeting 1301 ubiquitin-associated human genes with 6 sgRNAs per gene was designed – is it commercial or custom built?

We thank reviewer #1 for pointing out the need for clarification of the details of the UPS-focused library. To account for this critique, we clarified that our library is custom-made. Library design and generation are described in the *Methods* section. In addition, we provided the list of included genes and their respective sgRNAs in the Supplementary Table 3.

Changes to the manuscript:

- The relevant sentence in the *Methods* section now reads:

“A **custom-made** focused sgRNA library targeting 1301 ubiquitin-associated human genes with 6 sgRNAs per gene was designed based on the VBC score.”

- The list of genes and their targeting sgRNAs was added as a separate file: **Supplementary Table 3**. UPS-focused library.

5. There are errors in captions 3i) and 4e) for the expected mass of the FBXO22-SP3CHO adduct – please correct.

We thank reviewer #1 for bringing this error to our attention. We have adjusted the respective captions.

Changes to the manuscript:

- **Page 9, line 4:** “Expected mass for FBXO22-SP3CHO adduct: **45374 Da.**”
- **Page 10, line 17:** “Expected mass for FBXO22-SP3CHO adduct: **45374 Da.**”

6. Page 9, like 20 says “all three” assayed degraders, but you reference more than three in the first sentence. Please make it clear that “all three” does not refer to the BRD4-targeting compounds (since it did not work).

We appreciate this correction and have adjusted the respective sentence for clarification.

Changes to the manuscript:

- **Page 9, lines 41-42 (changes highlighted):** “In contrast, reconstitution with FBXO22-WT re-sensitized KO cells to target destabilization by **the NSD2, XIAP and FKBP12 targeting degraders.**”

7. Figure 2D shows proof of recruitment of FBXO22 for degradation by coimmunoprecipitation. The FKBP12 is a double band in the IP:HA and a single band in the Input. Only the bottom band shows rescue from SP3N degradation by carfilzomib. Can you explain the two bands and what the upper band represents? Non-specific antibody result? Degradation product?

We thank reviewer #1 for bringing attention to the presence of the upper band observed in the co-immunoprecipitation lane. In all IP experiments we have included appropriate negative controls where we expressed each tagged protein alone, as well as the empty parental vector of FKBP12 that does not express any protein. Notably, we observed the presence of the upper band in these controls as well (Reviewer-only Fig. 2). Based on these findings, we have concluded that the upper band represents an unspecific antibody band. For clarification, we added the note “unspecific band” in the figure caption.

Reviewer-only figure 2. Annotation of unspecific band.

Co-immunoprecipitation of 2HA-FBXO22 and Nluc-3xFlag-FKBP12 following treatment with DMSO, 1 µM dFKBP1, 1 µM or 10 µM SP3N or 10 µM SP3NAc for 4 h in the presence of 1 µM carfilzomib. Anti-HA beads were used for the FBXO22 enrichment, and anti-HA or anti-Flag antibodies for FKBP12 and FBXO22 protein levels in the input and IP fractions. Samples without carfilzomib were used as controls for degradation. Conditions with only Nluc-3xFlag-FKBP12, only 2HA-FBXO22 or empty vector were used as negative controls for the IPs. IP: immunoprecipitation, IB: immunoblot. *unspecific band. Representative image of n=3 independent experiments.

Changes to the manuscript:

- The Figure 2d caption (page 6, line 19) now includes: “*unspecific band”

8. Figure 3, the work regarding quantification around conversion of SP3N to SP3CHO with and without cells (Figure 3b, c, d, e). I would encourage the authors to edit the figure to data critical to support quantification levels. The figure needs to be understandable without significant deep-dive into the text – simplify.

We thank reviewer #1 for their comment. To address this point, we removed the plot illustrating the quantification of SP3N and SP3CHO in media without cells, thus focusing panels 3b, 3c and 3d solely on *cellular* assays. In addition, to simplify the data presentation, we changed the graph type to superimposed bar graph format, facilitating the clearer identification of the FCS-dependent SP3N-to-SP3CHO conversion.

Changes to the manuscript:

A modified version of Fig. 3. has been added (Page 8):

Fig. 3c Quantification of SP3N and SP3CHO (pmol) using UPLC-MS/MS, in KBM7 iCas9 cells. 1 µM SP3N was added in IMDM + 10% FCS or Opti-MEM - FCS and incubated at 37 °C for 6 h. Mean ± s.d of n=3 technical replicates.

- The respective paragraph in the text now reads (**page 7, lines 10-14, changes highlighted**):
“To confirm the presence of the SP3N-derived aldehyde (SP3CHO) and the dependence on FCS for this metabolic step, we **treated KBM7 cells with SP3N** in IMDM + 10% FCS or Opti-MEM without FCS and used ultra-performance liquid chromatography-mass spectrometry (UPLC-MS/MS) to detect the formation of the aldehyde species. Our results reveal the detection of SP3CHO at **6 h** of incubation, and only in conditions containing FCS (Fig. 3c).”

Reviewer #2 (Remarks to the Author):

In this paper, the authors describe the identification of compound SP3N (SLF-PEG3-NH₂) that is capable of recruiting the SCF-FBXO22 ligase to induce the polyubiquitylation of FKBP12, leading to its degradation by the proteasome, and further unravel the mechanism of action of SP3N. They nicely dissect the serendipitous discovery that SP3N functions as a prodrug and tie their results back to previously disclosed compounds (UNC 5153 & GNE compound 10). The authors' findings show a higher potency of degradation compared to previous studies on covalent degraders, which strengthens the case for further exploration of FBXO22 as a target for drug discovery. I anticipate that the TPD community (and scientific community at large) will be very enthusiastic about this study. Overall, after appropriate revisions, I believe that the paper would be a suitable fit for publication in Nature Communications.

We thank reviewer #2 for their positive remarks and insightful suggestions.

Minor comments:

1. Please add statistical analysis to all the main and supplementary figures.

We thank reviewer #2 for bringing this to our attention, we have adapted the manuscript accordingly.

Changes to the manuscript:

- We modified the captions of all figures 1-4 and supplementary figures 1-5 to include the statistical analyses.

2. Discussion section

a. Please discuss in more details if there are any potential advantages (such as tissue expression and selectivity, disease specificity, unique binding sites etc.) of harnessing FBXO22 compared with the currently available E3 ligases. For example, it would be nice to include a supplemental figure that displays the expression profile of FBXO22 in normal and disease (e.g. GTEx and TCGA) compared to CRBN.

We thank reviewer #2 for their recommendation. We adjusted the *Discussion* section to highlight the potential advantage of using FBXO22 over the commonly used CRBN or VHL, to overcome the emerging tumor resistances observed with the current pre-clinical or clinical CRBN- or VHL-based degraders. In addition, we included a new Supplementary Fig. 5 comparing the FBXO22 (Supplementary Fig. 5d) and CRBN (Supplementary Fig. 5e) expression data between tumor and healthy tissues, using GEPIA2, a tool for analyzing the RNA sequencing expression data of 9,736 tumors and 8,587 normal samples from the TCGA and the GTEx projects (Tang et al., *Nucleic Acids Res.* 2019).

Changes to the manuscript:

- A new supplementary Fig. 5 has been added (see supplementary information):

Supplementary Fig. 5 d, e Comparisons of the median gene expression (TPM; transcripts per million) of FBXO22 (d) or CRBN (e) in different cancer tissues and the respective normal tissues extracted from TCGA and GTEx using GEPIA2. The gray bars represent the TPM in the normal tissues, whereas the red bars represent the TPM in cancer tissues. The median FBXO22 or CRBN levels among all healthy tissues is represented by the dotted horizontal line. ACC, Adrenocortical carcinoma; BLCA, Bladder Urothelial Carcinoma; BRCA, Breast invasive carcinoma; CESC, Cervical squamous cell carcinoma and endocervical adenocarcinoma; CHOL, Cholangio carcinoma; COAD, Colon adenocarcinoma; DLBC, Lymphoid Neoplasm Diffuse Large B-cell Lymphoma; ESCA, Esophageal carcinoma; GBM, Glioblastoma multiforme; HNSC, Head and Neck squamous cell carcinoma; KICH, Kidney Chromophobe; KIRC, Kidney renal clear cell carcinoma; KIRP, Kidney renal papillary cell carcinoma; LAML, Acute Myeloid Leukemia; LGG, Brain Lower Grade Glioma; LIHC, Liver hepatocellular carcinoma; LUAD, Lung adenocarcinoma; LUSC, Lung squamous cell carcinoma; OV, Ovarian serous cystadenocarcinoma; PAAD Pancreatic adenocarcinoma; PCPG, Pheochromocytoma and Paraganglioma; PRAD, Prostate adenocarcinoma; READ, Rectum adenocarcinoma; SARC, Sarcoma; SKCM, Skin Cutaneous Melanoma; STAD, Stomach adenocarcinoma; TGCT, Testicular Germ Cell Tumors; THCA, Thyroid carcinoma; THYM, Thymoma; UCEC, Uterine Corpus Endometrial Carcinoma; UCS, Uterine Carcinosarcoma.

- The respective sentence in the discussion section now reads (**page 11, lines 39-41**):

“This is further supported by TCGA data that indicates the elevated expression levels in tumor tissue as a differentiating characteristic compared to CRBN, the most-frequently pursued E3 ligase for TPD applications (Supplementary Fig. 5d, e).”

- The relevant paragraph in the **Methods** section was added.

b. The authors comment that native substrates have been mapped for FBXO22. Do the authors anticipate if conjugation to C326 would disrupt native substrate binding? If so, are there any toxicity = implications?

This is an interesting point. We investigated the potential impact of SP3N treatment on native substrates, by looking at our whole proteome experiment. We observed that the levels of several literature-reported FBXO22 substrates, such as BACH1, PTEN, KLF4 or KDM4B (Lignitto et al., *Cell*. 2019; Ge et al., *Nat Commun*. 2020; Tian et al., *Oncotarget*. 2015; Johmura et al., *J Clin Invest*. 2018) remained unchanged upon treatment with SP3N, suggesting that the conjugation to C326 likely does not disrupt native substrate binding (Reviewer-only Fig. 3a). In fact, not a single protein was significantly stabilized in that experiment, meaning that also putative non-annotated substrates are not subject to stabilization due to pharmacologic FBXO22 hijacking. Furthermore, we did not observe any differences in toxicity profiles among the different FBXO22 cellular backgrounds upon treatment with SP3N or SP3CHO (Reviewer-only Fig. 3b). These results support the conclusion that these treatments do not significantly interfere with endogenous substrates and do not pose apparent toxicity implications.

Reviewer-only figure 3: SP3N impact on endogenous FBXO22 substrates and toxicity implications.

a Whole proteome analysis using tandem mass tag quantification in HEK293T cells treated with DMSO or 1 μ M SP3N for 16 h. Log₂ fold-changes (Log₂FC) and $-\log_{10}$ -transformed Benjamini–Hochberg adjusted one-way analysis of variance (ANOVA) *P* value compared with DMSO treatment. Hits with Log₂FC < -1 and $-\log_{10}$ (adjusted *P* value) > 2 are annotated as downregulated (red) and hits with Log₂FC > 1 and $-\log_{10}$ (adjusted *P* value) > 2 are annotated as upregulated (blue). ns = non-significant. Data from n=3 replicates. **b** Dose-resolved, normalized viability of HEK293T FKBP12-BFP-P2A-mCherry WT, FBXO22 KO clone or FBXO22 KO clone reconstituted with 2HA-FBXO22-WT or 2HA-FBXO22-C326, treated with SP3N or SP3CHO. In gray highlighted the working concentrations. Mean \pm s.d. of n=3 technical replicates; representative of n=2 independent experiments.

Major comments:

1. In figure 3D, 3F (+FCS group experiments), the SP3CHO outperforms SP3N in the presence of FCS. Since it is stated by authors that FCS is not changing SP3CHO chemical properties, what is the reason for reduced activity? (permeability?)

We thank reviewer #2 for this insightful question regarding the observed difference in activity between SP3CHO and SP3N. While we don't have a definitive answer, it is plausible that SP3CHO exhibits higher potency due to its ability to bypass the dependence on amine oxidase activity. By already being in its active form, SP3CHO circumvents the need for conversion by amine oxidase enzymes, which may contribute to its enhanced potency compared to SP3N. Supporting this hypothesis, the metabolomics data on Fig. 3c show that only a small fraction of SP3N undergoes the metabolic conversion to the active aldehyde. This implies this metabolic conversion as rate-limiting step and suggests that the effective concentrations of active SP3CHO are different in the + FCS conditions. To bring this point to the attention of our readers, we have decided to address it with additional text in the *Results* section.

Changes to the manuscript:

- The relevant sentence added in the *Results* section (page 7, lines 28-30, changes highlighted):

"In cellular degradation assays, in the presence of FCS, both aldehyde species outperform their matched alkylamine analog (Fig. 3e, Supplementary Fig. 3b). This suggests that metabolic conversion might act as a rate-limiting step (Fig. 3c)."

2. In figure 3F, there is a significant hook effect observed at the highest tested dose for SP3CHO in - FCS group, however, this is not the case for +FCS group. i) Please explain why there is such a big hook in the absence of FCS for SP3CHO? Also, SP3CHO is a bit more potent in the absence of FCS, ii) Is this difference statistically significant? If yes, it is important to show how FCS affects the potency of the compound. For instance, does FCS change cellular permeability, cellular uptake, binding affinity, dissociation kinetics?

We thank reviewer #2 for pointing out the differences in SP3CHO potency in medium +/- FCS. To facilitate the direct comparison of the SP3CHO potency, we overlaid the degradation effects of SP3CHO in media +/- FCS (and the respective SP3N conditions). We indeed observe a slightly increased potency of SP3CHO and a more pronounced hook effect in the condition - FCS (Reviewer-only Fig. 4a, b).

Reviewer-only Fig. 4. Comparison of SP3N & SP3CHO potency in media +/- FCS.

a, b Flow-cytometry based degradation assay in KBM7 iCas9-FKBP12-BFP-P2A-mCherry reporter cells washed to remove FCS, resuspended in IMDM + 10% FCS or Opti-MEM - FCS and treated with the indicated concentrations of DMSO, SP3N (a) or SP3CHO (b) for 6 h. The BFP/mCherry ratio was normalized to DMSO. Mean \pm s.d of n=3 biological replicates.

Following the reviewer's suggestion, we decided to measure the intracellular SP3CHO levels after cellular incubation in media +/- FCS. We decided to focus on an early timepoint since we anticipate that at prolonged exposure, SP3CHO would react with non-FBXO22 cysteine residues (see ABPP experiment below) and therefore escape our quantification. Note that this experiment was conducted in FBXO22 KO cells to have control over adduction of the FBXO22 cysteine.

Indeed, we could show that SP3CHO levels are significantly lower (>60% drop) in cells that were treated in FCS-containing media. This suggest that FCS negatively affects the available intracellular SP3CHO levels, presumably due to unspecific BSA binding.

Changes to the manuscript:

- A revised Supplementary Fig. 3c panel was added (see supplementary information):

Supplementary Fig. 3c Quantification of intracellular SP3CHO (pmol) in medium +/- FCS. KBM7 iCas9 FKBP12-BFP-P2A-mCherry FBXO22 KO cells were treated with 1 μ M SP3CHO in OptiMEM +/- FCS for

5 min and the SP3CHO levels were quantified using UPLC-MS/MS. Mean \pm s.d. of $n=4$ technical replicates.”

- The respective sentence in the **Results** section now reads (page 7, lines 32-36):

“Interestingly, SP3CHO is slightly more potent and shows a more pronounced hook effect (a phenomenon typically observed with PROTACs) in the absence of FCS (Fig. 3e). UPLC-MS/MS based, targeted quantification of SP3CHO levels after short cellular treatment revealed significantly elevated levels of SP3CHO in cells treated in media lacking FCS as a plausible mechanism for this differential potency (Supplementary Fig. 3c).”

3. Based on supplementary Fig. 3e. (FKBP degradation), the authors argue that aldehydes are favored over other electrophiles. This statement seems overly broad with little supporting evidence. For example, the preprint from Xiaoyu Zhang used a chloroacetamide-based degrader and observed comparable degradation potencies. It is likely that differences in warhead or linker are a driving factor regulating potency. To make a stronger case, the authors should (at the least) compare the percent FBXO22 adduct formation between the compounds listed in Fig. S3e to ensure that differences aren't simply due to reactivity.

We thank reviewer #2 for their remark. We followed their suggestion and performed intact-MS to explore the potential adduct formation with the SP2-Cl and SP2-Acry. As positive control for adduct formation we used the FBXO22-SP3CHO adduct. No additional peaks were observed in the FBXO22 + SP2-Cl or FBXO22 + SP2-Acry, supporting that these covalent warheads do not bind to FBXO22 (see Reviewer-only Fig. 5). We do not rule out the possibility that other covalent warheads could bind to FBXO22, as supported by the preprint highlighted by reviewer #2. However, it appears that various cysteine residues of FBXO22 may be preferred by different covalent molecules. Further investigations and structural elucidation of FBXO22 are necessary to pinpoint the specific determinants of these compounds' binding, but are outside the scope of this manuscript. To further highlight that FBXO22 can also be co-opted with other electrophilic warheads, we have added additional text into the discussion.

Reviewer-only Fig. 5. SP2-Cl and SP2-Acry do not form adducts with SKP1-FBXO22.

Intact mass spectrometry for the identification of FBXO22-SP2-Cl or SP2-Acry complex formation. 20 μ M FBXO22-SKP1 complex was incubated with 100 μ M SP3CHO, SP2-Cl or SP2-Acry for 10 min and analyzed with LC-MS. The spectra of FBXO22, FBXO22 + SP3CHO, FBXO22 + SP2-Cl or FBXO22 + SP2-Acry are shown. Expected mass for FBXO22: 44649 Da. Expected mass for FBXO22 + SP3CHO adduct: 45374 Da. Expected mass for FBXO22 + SP2-Cl adduct: 45372 Da. Expected mass for FBXO22 + SP2-Acry adduct: 45386 Da.

Changes to the manuscript:

- The added sentence in the *Discussion* section (page 12, lines 3-5, changes highlighted):

“This is of particular interest in light of recent findings by Basu et al., highlighting that additional cysteine residues (C227, C228) of FBXO22 are in principle ligandable with chloroacetamide-based compounds.³⁷”

4. The authors should run proteome-wide APBB-MA experiment with their degraders. This information would provide a deeper understanding of compound selectivity and general activity of FBXO22 (e.g. percent FBXO22 engagement necessary for TPD).

We thank reviewer #2 for their valuable feedback. We followed their suggestion and performed a proteome-wide TMT-ABPP experiment with SP3CHO. Our results indicate approximately 30% engagement of the FBXO22-Cys326 upon SP3CHO treatment of HEK293T lysates (Supplementary Fig. 4g). Interestingly, our compound exhibited relatively low reactivity, with only a few additional targets being significantly engaged, none exceeding 20-30% engagement. Of note, none of the other quantified FBXO22 cysteines showed any engagement (Supplementary Fig. 4h). These findings provide valuable insights into the selectivity and activity of our compound and confirm the direct and selective engagement of FBXO22-C326 over other FBXO22 cysteine residues. It's worth noting that to enhance the visibility of FBXO22 Cys in the ABPP experiment, we had to spike in recombinant SKP1-FBXO22 protein. This was necessary as hundreds of previous experiments have shown that many relevant FBXO22 peptides are otherwise not quantified (presumably due to low abundance).

Changes in the manuscript:

- A modified Supplementary Fig. 4. has been added (see supplementary information):

Supplementary Fig. 4 g Proteome-wide TMT-ABPP profiling of SP3CHO in HEK293T cell lysates spiked with 0.15 μ g recombinant SKP1-FBXO22 and treated with 40 μ M SP3CHO for 1.5 h. More than 20,000 cysteine sites were quantified. Log2 fold-changes (Log2FC) were calculated based on the DMSO treated cells. Red dots represent cysteine sites with Log2FC < -0.45 and -logPvalue > 2. Data from n=3 replicates. **h** TMT relative abundances (RA) of the quantified FBXO22 cysteines in the TMT-ABPP as described in g. RA are normalized to DMSO. Mean \pm s.d. of n=3 replicates.

- The relevant paragraph in the manuscript describing these results now reads (page 9, lines 28-33):

“To confirm the proteome-wide selective engagement by SP3CHO, we performed global reactive cysteine profiling in HEK293T cell lysates by TMT-ABPP.^{43,44} This revealed that approximately 30% of FBXO22-C326 was engaged by SP3CHO. No other FBXO22 Cys residues were detected as engaging SP3CHO (Supplementary Fig. 4g, h). Notably, SP3CHO generally exhibited low reactivity with 5 other Cys (HDAC1-C100, GPX4-C102, TARS1-C254, PSMB1-C82;89 and PPAT-C503).”

- The **Methods** section of the manuscript has been updated to include the aforementioned experiment.

5. As stated by the authors, there are limits to the generalizability of the approach. However, since proteins are not equally amenable to E3-based degraders and as BRD4 is the only protein tested for this system in this paper, i) It would be nice to test other systems where many targets can be targeted such as using pan-kinase inhibitors as warhead domains. ii) It is also possible that the compound forms a ternary complex (TCF) with BRD4 and FBXO22 but unable to form a favorable TCF and/or access available lysine for ubiquitination. So, it is crucial for this study to at least show whether the generated BRD4 series can bind to FBXO22 (Binary complex formation), and if ternary complex can be formed. iii) Based on the data provided in supplementary Fig. 4b, authors tested BRD4 degradation using BRD4(short)-BFP-P2A-mCherry cells. Is this possible that fusing BRD4 to BFA restricting favorable ternary complex formation and lysine accessibility for E3 ligase FBXO22 and the ubiquitination machinery? Please assess the degradation potency of the BRD4 library against endogenous BRD4 using other systems such as HiBiT tagging systems, or Lumit or immunoblots?

We appreciate the depth of suggestions and comments raised by reviewer #2. We have structured our response to the individual subsections in the following.

i) We agree that further exploring the target scope of this new TPD strategy would be very intriguing. We also think that multi-targeted kinase inhibitors would be an excellent choice since it would also allow benchmarking against standard PROTAC/E3 ligases (CRBN, VHL) by comparing to existing data (for instance, as reported in the “degradable kinome” paper by laboratories of Gray and Fischer). However, we would argue that this broader survey is best done in a separate and dedicated follow-up study. The emphasis of this manuscript is really on the dissection of the underlying mechanism of action and the concluding notion that the identified mechanism is not a singular “lucky shot” but already extends over other target protein classes (the histone methyltransferase NSD2 and the E3 ligase XIAP). To go beyond this and into a systematic survey is going to be very resource (chemistry- and proteomics-) heavy and, in combination with the additional follow-up and validation experiments, would not be compatible with the timeframe of our revision work. We hope that reviewer #2 will understand our concerns.

ii) We thank reviewer #2 for their feedback and for emphasizing the importance of investigating potential ternary complex formation between FBXO22, BRD4, and JQ1-alkylamines. In response to this suggestion, we conducted co-immunoprecipitation experiments with 2HA-FBXO22 and BRD4short-3V5 upon treatment with the different JQ1-alkylamines (Supplementary Fig. 5c). Our results did not reveal any evidence of complex formation between FBXO22 and BRD4 upon treatment with the JQ1 analogues. It is noteworthy that we did observe complex formation between CRBN-dBet6-BRD4 or FBXO22-SP3N-FKBP12, which served as controls for this experimental setup.

iii) We understand the concern regarding the potential interference caused by the fusion of BRD4short to BFP in restricting favorable ternary complex formation and lysine accessibility. In response to reviewer’s #2 suggestion, we explored the degradation potency of the JQ1-alkylamines against the endogenous BRD4 protein using western blot analysis (Supplementary Fig. 5b). Our results confirmed that the JQ1 analogues do not degrade the endogenous BRD4. Notably, the alkylamines did also not induce BRD3 degradation, but did reduce c-Myc levels as a response to BRD4 inhibition, similar to the effects of JQ1 treatment. We believe that this is an important observation as it highlights that the compounds are permeable and reach intracellular concentrations that are sufficient for target (BRD4) occupancy.

Changes to the manuscript:

- A new Supplementary Fig. 5 was added (**see supplementary information**):

Supplementary Fig. 5. Expandability of co-opting FBXO22.

B Immunoblot of endogenous BRD4, BDR3 and c-Myc in KBM7 iCas9 FKBP12-BFP-P2A-mCherry cells treated with DMSO, 1 μ M dBet6, 1 or 10 μ M JQ1 and 1 μ M or 10 μ M of the JQ1-alkylamines: JQ1-C6-NH₂, JQ1-C8-NH₂, JQ1-PEG2-NH₂, JQ1-PEG3-NH₂ or JQ1-PEG4-NH₂ for 16 h. GADPH is the loading control. Representative blot of n=2 experiments. **c** Co-immunoprecipitation of 2HA-FBXO22 and BRD4s-3xV5 following treatment with DMSO, 1 μ M dBet6, 10 μ M JQ1 or 10 μ M of any of the JQ1-alkylamines (JQ1-C6/C8/PEG2/PEG3 or PEG4-NH₂), for 4 h in the presence of 1 μ M carfilzomib. Co-immunoprecipitation of 2HA-FBXO22 with Nluc-3xFlag-FKBP12 upon treatment with 10 μ M SP3N or of 2HA-CRBN with BRD4s-3xV5 upon treatment with 1 μ M dBet6 were used as positive control conditions. Cells transfected with only HA-empty vector or V5-empty vector were used as negative controls for the IPs. Anti-HA beads were used for the FBXO22 or CRBN enrichment, anti-Flag or anti-V5 antibodies were used to quantify the FKBP12 and BRD4 protein levels, while anti-CRBN or anti-FBXO22 were used for either the 2HA-tagged or endogenous CRBN and FBXO22 proteins. Representative blot of n=2 experiments. IP: immunoprecipitation, IB: immunoblot.

The relevant paragraph in the **Results** section describing these results has been added (**page 9, lines 46-52, changes highlighted**):

“Interestingly, a set of alkylamine-based analogues building off the BET-bromodomain inhibitor JQ1 did not degrade BRD4, a target that is frequently utilized as proof of concept for prototypical degraders that co-opt novel E3 ligases. While immunoblot analysis of the BRD4 transcriptional target MYC implies cellular target engagement of this set of analogues, co-IP experiments reveal a lack of ternary complex formation as a likely reason for the observed lack of degradation (Supplementary Fig. 5b, c). In sum, these data suggest that the concept of alkylamine-base degraders is generalizable, yet will require optimization on a target-by-target level.”

6. In the discussion, the authors make an interesting argument that the prodrug strategy could be exploited to drive tumor restricted degradation by targeting indications where amine oxidases are upregulated. Have the authors tested cell lines that express high vs low to see if selective degradation is observed? This data would strengthen the paper.

Given the extracellular nature of amine oxidases, we recognize that cell lines may not provide the most suitable model for directly assessing their impact on our “prodrug” strategy. Future studies should be exploring alternative experimental approaches that better reflect the physiological context in which amine oxidases function. One potential avenue would be use of *ex vivo* or *in vivo* models that better mimic the tumor microenvironment and allow for the assessment of tumor-restricted degradation in a more relevant context. By utilizing such models, we can evaluate the interplay between amine oxidase activity, prodrug activation and tumor-specific degradation, providing a more accurate representation of the potential therapeutic utility of our approach.

Changes to the manuscript:

A sentence was added in the **Discussion** section (**page 11, lines 45-49**):

“Future studies should explore experimental approaches that better reflect the physiological context in which amine oxidases function. One potential avenue would be the use of *ex vivo* or

in vivo models that better mimic the tumor microenvironment and allow for the assessment of tumor-restricted amine-to-aldehyde conversion and target degradation.”

7. The mutagenesis experiment showing that C326 is the target cysteine for their degraders is compelling. To provide even stronger evidence, the authors should consider showing that this mutant still engages Skp1 in cells. This would rule out artifacts due to changes in protein folding.

We understand reviewer’s #2 concern regarding the potential effects of the C326A mutation on FBXO22 folding and SCF complex function. This concern was also independently raised by reviewer #1, therefore we put emphasis on addressing this comment with additional recombinant experiments and cellular assays. The experimental approaches we followed to address the reviewers’ comments are outlined extensively in our reply to the “major point 1” raised by reviewer #1.

To facilitate reviewer #2, we paste our response again here:

We performed nano differential scanning fluorimetry (NanoDSF) to test the protein stability of the SKP1-FBXO22-WT and SKP1-FBXO22-C326A complexes. These experiments revealed thermal stability of both variants, suggesting that C326A does not affect protein folding (Supplementary Fig. 4c).

Changes to the manuscript:

- The revised Supplementary Fig. 4c panel has been added (see supplementary information):

c

Supplementary Fig. 4c Normalized fluorescence data from nano differential scanning fluorimetry (NanoDSF) of SKP1-FBXO22-WT and SKP1-FBXO22-C326A. Heat denaturation was measured at a rate of 1 °C/min. Measurements were performed in duplicates.

In addition, we added experimental evidence that the FBXO22-C326A mutant incorporates into a functional CRL complex. CRL ligases are actively disassembled in cells via a process that is dependent on cullin deneddylation via the COP9 signalosome (CSN). Pharmacologic CSN inhibition thus arrests a CRL in a constitutive active state that frequently leads to auto-degradation (particularly on the substrate receptor). Hence, induction of protein degradation of a substrate receptor upon CSN inhibition strongly implies that it is incorporated in an active CRL (Schlierf et al, *Nat Commun*, 2016; Mayor-Ruiz et. al, *Mol Cell*, 2019; Henneberg et. al., *Nat Chem Biol*, 2023). Supporting the observation that the C326A mutation does not affect its functionality, we observed that cellular treatment with the CSN inhibitor CSN5i-03 led to similar degradation profiles of both FBXO22 variants (Supplementary Fig. 4d).

Changes to the manuscript:

- The revised Supplementary Fig. 4d panel has been added (see supplementary information):

d

Supplementary Fig. 4d Immunoblot of 2HA-FBXO22-WT or 2HA-FBXO22-C326A in HEK293T-FKBP12-BFP-P2A-mCherry FBXO22 KO single clone transduced with 2HA-FBXO22-WT or -C326A cDNAs. Cells were treated with DMSO or 500 nM CSN5i-03 for 1-16 h. α Tubulin is the loading control. Representative plot of $n=2$ independent experiments.

Finally, these findings were further supported by an *in vitro* autoubiquitylation assay which revealed comparable rates of autoubiquitylation of FBXO22-WT and FBXO22-C326A variants, thus again suggesting intact SCF complex activity (Supplementary Fig. 4e).

Changes to the manuscript:

- The revised Supplementary Fig. 4e panel has been added (see supplementary information):

e

Supplementary Fig. 4e Pulse-chase *in vitro* autoubiquitylation assay of FBXO22 with SKP1-FBXO22-WT and SKP1-FBXO22-C326A. UB*: fluorescent ubiquitin. Representative blot from $n=2$ independent experiments.”

- The relevant paragraph in the manuscript describing the results for all Supplementary Fig. 4c, d, e now reads (page 9, lines 13-20):

“To rule out potential deleterious effects of the C326A mutation on FBXO22, we turned to Nano differential scanning fluorimetry (NanoDSF) which revealed thermal stability (Supplementary Fig. 4c). In addition, FBXO22 WT and C326A are similarly sensitive to pharmacologically induced auto-degradation via COP9 signalosome inhibition (Supplementary Fig. 4d), supporting the notion that the mutant incorporates into a functional SCF complex also in cells.⁴⁰⁻⁴² Further, an *in vitro* autoubiquitylation assay demonstrated comparable autoubiquitylation of recombinant FBXO22-WT and FBXO22-C326A variants, hence again suggesting intact SCF complex activity (Supplementary Fig. 4e).

- The **Methods** section of the manuscript has been updated to include the aforementioned experiments.

Reviewer #3 (Remarks to the Author):

In the manuscript 'Alkylamine-tethered molecules recruit FBXO22 for targeted protein degradation'

Winter and colleagues identify and rigorously characterise a novel degrader prodrug SP3N that consists of a known binder of FKBP12 appended with a flexibly alkylamine tail. In cells the molecule is metabolized to an active aldehyde that recruits the SCF/FBXO22 ligase through covalent adduction of Cys326 of FBXO22. The authors further demonstrate that their results are in line with a similarly detailed recent study by the Arrowsmith lab that described the same mechanism when investigating alkyl amine-based degraders of NSD2 and XIAP. Whilst the data on the FKBP12 degrader suggest broader applicability of the approach absence of degradation of alkyl amine tethered analogs of the BET inhibitor JQ1 suggest that there are limits to the generalisability of the described concept.

The study is comprehensive, data quality is very good, claims are generally well grounded on data and the observation that alkylamine thethers provide prodrugs leading to reversible covalent linkage to FBXO22 and proximity induced degradation of target proteins is highly interesting and relevant.

My only concern is that this study closely follows work published as a preprint by the Arrowsmith lab and the incremental novelty is very limited.

We thank reviewer #3 for their positive feedback regarding the quality of our study. We acknowledge the reviewer's point regarding similarities of our study and the work published as a preprint by the Arrowsmith lab. It's noteworthy that both studies feature different compounds (our study describing new chemical matter) as well as different discovery approaches and experimental strategies geared to uncover the underpinning mechanism of action. Nevertheless, both studies arrived at alkylamine-mediated FBXO22 recruitment as a unifying mechanism. This convergence of findings underscores the robustness of the novel mechanism uncovered, and further validates its significance. This aspect is particularly noteworthy in highlighting FBXO22 as a novel E3 within the TPD field. Both studies independently support the potential generalizability of this mechanism, offering insights into its broader applicability and relevance. We believe this strengthens the overall impact of both manuscripts and underscores the importance of the identified mechanism in the field.

REVIEWERS' COMMENTS

Reviewer #1 (Remarks to the Author):

Very thorough revision, excellent job! All reviewer concerns addressed to satisfaction, no further comments.

Manuscript recommended for publication.

Reviewer #2 (Remarks to the Author):

The authors have done a nice job in addressing my comments and concerns. I have no further constructive criticism or feedback to offer. Therefore, I recommend that Nature proceeds with publishing this article.